# Stochastic Optimization with Random Search

## Abstract

We revisit random search for stochastic optimization, where only noisy function evaluations are available. We show that the method works under weaker smoothness assumptions than previously considered, and that stronger assumptions enable improved guarantees. In the finite-sum setting, we design a variance-reduced variant that leverages multiple samples to accelerate convergence. Our analysis relies on a simple translation invariance property, which provides a principled way to balance noise and reduce variance.

## 1. Introduction

Zeroth-order, also known as derivative-free, optimization studies the minimization of an objective using only function evaluations, which may be noisy. Such methods are indispensable when gradients are unavailable, unreliable, or prohibitively expensive to obtain, as in black-box adversarial attacks (Chen et al., 2017; Ughi et al., 2022), hyperparameter tuning (Koch et al., 2018; Turner et al., 2021), reinforcement learning through rollouts (Mania et al., 2018; Malik et al., 2020), and simulation-based optimization (Conn et al., 2009).

Two broad paradigms dominate the zeroth-order literature. *Gradient-approximation* methods estimate directional derivatives from finite differences, typically along random directions, and combine them to form an approximate descent direction, then apply first-order-style updates (Nesterov & Spokoiny, 2017; Ghadimi & Lan, 2013; Liu et al., 2018). While these approaches achieve strong oracle complexities, they inherently rely on a smoothing or differencing step that introduces bias and typically requires careful tuning of the smoothing parameter and stepsize, which can substantially affect performance.

In contrast, *direct-search* methods operate purely by com-

paring function values at a small set of trial points and updating the current iterate whenever an improvement is found (Bergou et al., 2020; Golovin et al., 2020). They avoid explicit derivative estimation, are simple to implement, and are often robust when evaluations are noisy.

The *Stochastic Three Points* (STP) method of (Bergou et al., 2020) is one of the simplest and most efficient forms of direct-search methods, requiring only three function evaluations per iteration and incorporating random search directions (this is where random search comes from). It evaluates the objective at the current iterate and at two origin-symmetric perturbations along a random direction, then selects the best of the three. In the deterministic setting, STP achieves optimal complexity bounds: $O(d/\varepsilon^2)$ in the smooth non-convex case (Bergou et al., 2020), matching the best-known rate achieved by gradient estimation methods, and $O(d/\varepsilon)$ in the smooth convex case (El Bakkali & Saadi, 2025), matching the best known complexity bound among gradient estimation methods that do not use momentum.

Despite their appeal, STP and random search methods have been analyzed almost exclusively in the deterministic setting, where access to the full objective function is assumed. However, this assumption is often unrealistic in large-scale machine learning, where the objective typically involves a finite sum over millions of data points. Large-scale learning tasks such as deep reinforcement learning or empirical risk minimization require stochastic algorithms. While stochastic gradient estimation methods are well studied, stochastic random search methods remain poorly understood. Yet, their simplicity and freedom from discretization bias make them an attractive alternative to gradient-based approaches. This gap in the theory motivates the development of stochastic variants of random search that can operate effectively with noisy or subsampled objective estimates. Providing sharp guarantees for such methods is therefore both theoretically and practically important. To address this limitation, in (Boucherouite et al., 2024), they proposed *Minibatch STP* (MiSTP), which replaces the true objective with a minibatch estimate during the three-point comparison of standard STP. Their analysis, however, yields a suboptimal $O(d^3/\varepsilon^6)$ complexity under strong individual smoothness assumptions, significantly worse than the $O(d/\varepsilon^4)$ rate achieved by gradient-estimation methods like RSGF (Ghadimi & Lan, 2013). The goal of this work is to close this gap.

[1]Anonymous Institution, Anonymous City, Anonymous Region, Anonymous Country. Correspondence to: Anonymous Author <anon.email@domain.com>.

Preliminary work. Under review by the International Conference on Machine Learning (ICML). Do not distribute.

## 2. Related Work

**Direct search and STP.** Classical direct-search methods (see (Conn et al., 2009; Vicente, 2013)) rely on deterministic search directions and tend to scale poorly with the dimensionality. In contrast, the STP algorithm achieves improved complexity bounds: $\mathcal{O}(d/\epsilon^2)$ in the smooth non-convex setting (Bergou et al., 2020), and $\mathcal{O}(d/\epsilon)$ in the smooth convex setting (El Bakkali & Saadi, 2025). These results improve upon the $\mathcal{O}(d^2/\epsilon^2)$ and $\mathcal{O}(d^2/\epsilon)$ complexities obtained for deterministic direct-search methods (Vicente, 2013; Konečnỳ & Richtárik, 2014). Follow-up work extended STP with importance sampling (Bibi et al., 2020) and momentum (Gorbunov et al., 2020), but these analyses were conducted only for deterministic objective functions.

**STP in the stochastic setting.** To the best of our knowledge, the only prior attempt to analyze STP for stochastic objectives is due to Boucherouite et al. (Boucherouite et al., 2024). Their minibatch STP method achieves a complexity bound of order $\mathcal{O}(d^3/\varepsilon^6)$ under strong *individual smoothness* assumptions, requiring each stochastic component function $f_\xi$ to be smooth. This rate is substantially worse than the $\mathcal{O}(d/\varepsilon^4)$ complexity known for gradient-estimation-based methods such as RSGF (Ghadimi & Lan, 2013), which approximate gradients using finite differences.

**Gradient-estimation methods.** Zeroth-order stochastic methods based on gradient approximation (e.g., two-point estimators (Ghadimi & Lan, 2013), ZO-SVRG (Liu et al., 2018)) achieve a complexity bound of $\mathcal{O}(d/\varepsilon^4)$ under individual smoothness assumption. However, these methods introduce discretization bias and require careful tuning of smoothing parameters, which can significantly impact their empirical performance. In contrast, random search methods avoid discretization errors altogether but have historically lacked comparable theoretical guarantees in the stochastic setting.

**Our contribution in context.** Our contributions can be summarized as follows:

- **Individual smoothness.** We prove that Random Search achieves the optimal $\mathcal{O}(d/\varepsilon^4)$ complexity under the standard *individual smoothness* assumption, while retaining its bias-free and comparison-based update rule.
- **Average smoothness.** Under the weaker assumption that only the average function $f$ is $(L_0, L_1)$-smooth, we show that Random Search recovers the $\mathcal{O}(d^3/\varepsilon^6)$ rate previously obtained in (Boucherouite et al., 2024), but under strictly milder requirements.
- **Finite-sum variance reduction.** For finite-sum objectives, we demonstrate that variance reduction can be realized without the memory overhead of storing snapshots as in ZO-SVRG (Liu et al., 2018). We

establish a complexity of $\mathcal{O}\left(\min\left\{\frac{d^{4/3}n^{2/3}}{\varepsilon^{8/3}}, \frac{dn}{\varepsilon^2}\right\}\right)$, which improves upon the deterministic rate whenever $n \gg d/\varepsilon^2$.

- **Helper or human feedback.** We extend Random Search to settings where only *inexact comparison feedback* is available, such as A/B testing (Kohavi et al., 2009) or RLHF (Christiano et al., 2017; Ouyang et al., 2022). We prove that convergence holds up to an accuracy floor of order $\mathcal{O}(\sqrt{d\delta})$, where $\delta$ quantifies the inexactness of the helper or human feedback.

Together, these results provide the first unified analysis of Random Search across stochastic, finite-sum, and human-in-the-loop settings.

## 3. General Algorithmic Framework

We consider the stochastic minimization problem

$$\min_{x \in \mathbb{R}^d} f(\boldsymbol{x}), \qquad f(\boldsymbol{x}) := \mathbb{E}_{\xi \sim \mathcal{P}}[f_\xi(\boldsymbol{x})],$$

where $(\Xi, \mathcal{F}, \mathcal{P})$ is a probability space and the mapping $F$ defined on $\mathbb{R}^d \times \Xi$ by $F(x, \xi) = f_\xi(\boldsymbol{x})$ satisfies: (i) $x \mapsto F(x, \xi)$ is differentiable for every $\xi \in \Xi$; (ii) $\xi \mapsto F(x, \xi)$ is $\mathcal{F}$-measurable for every $x \in \mathbb{R}^d$. We assume $f$ is finite-valued and bounded below. This framework includes the finite-sum setting common in machine learning: for $\Xi = \{1, \ldots, n\}$, uniform $\mathcal{P}$, and $F(x, i) = f_i(\boldsymbol{x})$, we recover $f(\boldsymbol{x}) = \frac{1}{n}\sum_{i=1}^n f_i(\boldsymbol{x})$. $\|\cdot\|$ stands for the $\ell_2$ norm.

### 3.1. Smoothness assumption

We adopt the following notion of smoothness, introduced in (Zhang et al., 2020).

**Assumption 3.1** $((L_0, L_1)$-smoothness)**.** A differentiable function $f : \mathbb{R}^d \to \mathbb{R}$ is $(L_0, L_1)$-smooth if, $\forall \boldsymbol{x}, \boldsymbol{y} \in \mathbb{R}^d$ :

$$\|\nabla f(\boldsymbol{x}) - \nabla f(\boldsymbol{y})\| \leq (L_0 + L_1\|\nabla f(\boldsymbol{x})\|)\|\boldsymbol{x} - \boldsymbol{y}\|. \tag{1}$$

This condition generalizes standard Lipschitz smoothness by allowing the gradient Lipschitz constant to grow proportionally with the gradient norm itself. It has been used to justify adaptive methods such as gradient clipping (Zhang et al., 2020).

We note that if a function $f : \mathbb{R}^d \to \mathbb{R}$ is $(L_0, L_1)$-smooth (i.e., satisfies Assumption 3.1), then for all $\boldsymbol{x}, \boldsymbol{y} \in \mathbb{R}^d$, we have:

$$f(\boldsymbol{y}) \leq f(\boldsymbol{x}) + \langle \nabla f(\boldsymbol{x}), \boldsymbol{y} - \boldsymbol{x} \rangle + \frac{L_0 + L_1\|\nabla f(\boldsymbol{x})\|}{2}$$
$$\times \|\boldsymbol{x} - \boldsymbol{y}\|^2. \tag{2}$$

### 3.2. Random directions

At each iteration the algorithm samples a random direction $s_t \sim \mathcal{D}$. We require the following assumption, adapted from (Bergou et al., 2020; Boucherouite et al., 2024).

**Assumption 3.2** (Exploration property of $\mathcal{D}$). The distribution $\mathcal{D}$ on $\mathbb{R}^d$ satisfies:

1. Normalization: $\mathbb{E}_{s \sim \mathcal{D}}[\|s\|^2] = 1$.

2. Exploration: there exists $\mu_{\mathcal{D}} > 0$ and a norm $\|\cdot\|$ such that for all $g \in \mathbb{R}^d$,

$$\mathbb{E}_{s \sim \mathcal{D}}\big[\,|\langle g, s \rangle|\,\big] \;\geq\; \mu_{\mathcal{D}} \|g\|. \tag{3}$$

This ensures that the distribution of search directions has sufficient coverage in expectation. In prior work, the norm $\|\cdot\|$ was allowed to depend on $\mathcal{D}$; here, by equivalence of norms in finite dimensions, we absorb this dependence into the constant $\mu_{\mathcal{D}}$.

In (Bergou et al., 2020, Lemma 3.4), the validity of Assumption 3.2 has been established for several distributions including:

1. For the normal distribution with zero mean and the identity matrix over $d$ as covariance matrix, i.e., $\mathcal{D} \sim N(0, \frac{I_d}{d})$:

$$\begin{cases} \mathbb{E}_{s \sim \mathcal{D}}[\|s\|^2] = 1, \\ \mathbb{E}_{s \sim \mathcal{D}}|\langle g, s \rangle| = \frac{\sqrt{2}}{\sqrt{d\pi}} \|g\|_2. \end{cases}$$

2. For the uniform distribution on the unit sphere in $\mathbb{R}^d$:

$$\begin{cases} \mathbb{E}_{s \sim \mathcal{D}}[\|s\|^2] = 1, \\ \mathbb{E}_{s \sim \mathcal{D}}|\langle g, s \rangle| \sim \frac{1}{\sqrt{2\pi d}} \|g\|_2. \end{cases}$$

### 3.3. Algorithm

**Stochastic Three Points (STP).** STP is a classical derivative-free randomized search method introduced in (Bergou et al., 2020). At iteration $t$, given an iterate $x_t \in \mathbb{R}^d$, a stepsize $\alpha_t > 0$, and a search direction distribution $\mathcal{D}$ on $\mathbb{R}^d$, the method draws $s_t \sim \mathcal{D}$ and forms the two symmetric trial points $x_t^+ = x_t + \alpha_t s_t$, $x_t^- = x_t - \alpha_t s_t$, then evaluates the objective at the three points $\{x_t^-, x_t, x_t^+\}$, and updates according to the best-of-three rule:

$$x_{t+1} \in \arg \min_{x \in \{x_t^-, x_t, x_t^+\}} f(x).$$

**Minibatch STP (MiSTP).** In the finite-sum setting, MiSTP (Boucherouite et al., 2024) adapts STP by replacing the full objective $f$ with a single empirical objective $f_{\mathcal{C}_t}$, computed from a minibatch $\mathcal{C}_t$, while retaining the three-point comparison: $x_{t+1} \in \arg \min_{x \in \{x_t^-, x_t, x_t^+\}} f_{\mathcal{C}_t}(x)$. Because the accept/reject decision is made using $f_{\mathcal{C}_t}$, the selected step may worsen the true objective $f$, and thus including the current point $x_t$ does not ensure anymore the monotonic improvement of the true objective as it is the case for classical STP. We drop this baseline evaluation at $x_t$ and adopt a two–point comparison:

$$x_{t+1} \in \arg \min_{x \in \{x_t^-, x_t^+\}} f_{\mathcal{C}_t}(x),$$

which evaluates both trial points on the same minibatch and avoids the noisy baseline at $x_t$. By denoting $M_t^+$ and $M_t^-$ the stochastic estimators of $f(x_t^+)$ and $f(x_t^-)$, we introduce the following algorithm:

---

**Algorithm 1** Mi2P: Stochastic Random Search (general form)

---

**Require:** initial point $x_0 \in \mathbb{R}^d$, step sizes $\{\eta_t\}_{t \geq 0}$, distribution $\mathcal{D}$
0: **for** $t = 0, 1, 2, \ldots$ **do**
0:  Sample direction $s_t \sim \mathcal{D}$
0:  Compute $x_t^+ = x_t + \eta_t s_t$ and $x_t^- = x_t - \eta_t s_t$
0:  Obtain stochastic estimates $M_t^+$ and $M_t^-$ of $f(x_t^+)$ and $f(x_t^-)$
0:  Update

$$x_{t+1} = x_t - \eta_t \, \text{sign}(M_t^+ - M_t^-)\, s_t$$

0: **end for**=0

---

This is a natural stochastic extension of the classical STP update: the method moves along $s_t$ in the direction suggested by the lower of the two estimated function values.

In general, the quantities $M_t^{\pm}$ may be constructed in any way that provides useful estimates of $f(x_t^{\pm})$; this includes subsampling-based evaluations (Section 4), variance-reduced finite-sum constructions (Section 5), and human or helper feedback (Section 6). More sophisticated mechanisms, such as momentum-based surrogates, also fit naturally into this framework.

### 3.4. General descent bound

The following lemma shows that, under Assumption 3.1, Algorithm 1 guarantees a descent step up to the stochastic approximation error.

**Lemma 3.3** (Descent bound). *Suppose $f$ is $(L_0, L_1)$-smooth and Assumption 3.2 holds. If $\eta_t \leq \mu_{\mathcal{D}}/L_1$, then*

$$\begin{aligned} f(x_{t+1}) \;\leq\; & f(x_t) - \mu_{\mathcal{D}} \eta_t \|\nabla f(x_t)\| + \frac{L_0}{2} \eta_t^2 \\ & + 2\,\mathbb{E}\big[\,|M_t^+ - f(x_t^+)| + |M_t^- - f(x_t^-)|\,\big]. \end{aligned} \tag{4}$$

The first three terms correspond to the smoothness-based descent typical in random search, while the last term captures the additional error introduced by the stochastic estimators $M_t^{\pm}$.

### 3.5. Translation invariance

A crucial property of Algorithm 1 is *translation invariance*: adding the same constant $c_t$ to both $M_t^+$ and $M_t^-$ does not change the outcome. This allows us to optimally shift the estimates and reduce the error term. Formally,

$$\inf_{c\in\mathbb{R}}\Big(|M_t^+ - c - f(\boldsymbol{x}_t^+)| + |M_t^- - c - f(\boldsymbol{x}_t^-)|\Big)$$
$$= \tfrac{1}{2}\left|(M_t^+ - M_t^-) - (f(\boldsymbol{x}_t^+) - f(\boldsymbol{x}_t^-))\right|. \quad (5)$$

Thus, rather than controlling each approximation error separately, it suffices to control the error in the *difference* $M_t^+ - M_t^-$, which is typically much smaller. This observation will be the key to our improved rates.

# 4. General Stochastic Functions with Subsampling

We consider the general stochastic optimization problem

$$f(\boldsymbol{x}) = \mathbb{E}_\xi[f_\xi(\boldsymbol{x})],$$

where only noisy evaluations of $f_\xi$ are available. To reduce noise, we estimate function values at the perturbed points $\boldsymbol{x}_t^{\pm} = \boldsymbol{x}_t \pm \eta_t \boldsymbol{s}_t$ using minibatches of size $b$:

$$M_t^{\pm} := \frac{1}{b}\sum_{j=1}^{b} f_{\xi_j}(\boldsymbol{x}_t^{\pm}), \qquad \{\xi_j\}_{j=1}^{b} \text{ i.i.d.}$$

We analyze two regimes of smoothness assumptions.

### 4.1. Case I: Weak average smoothness

We first assume that only the mean function $f$ satisfies $(L_0, L_1)$-smoothness (Assumption 3.1). To control the stochastic error, we assume a bounded variance of the component functions (cf. (Boucherouite et al., 2024)):

**Assumption 4.1** (Bounded variance). There exists $\sigma_0^2 < \infty$ such that for all $\boldsymbol{x} \in \mathbb{R}^d$, $\mathbb{E}[f_\xi(\boldsymbol{x})] = f(\boldsymbol{x})$ and $\mathbb{E}[(f_\xi(\boldsymbol{x}) - f(\boldsymbol{x}))^2] \le \sigma_0^2$.

Consequently, for $M = \frac{1}{b}\sum_{i=1}^{b} f_{\xi_i}(\boldsymbol{x})$, the average of $b$ i.i.d realizations of $f_\xi(\boldsymbol{x})$, we have: $\mathbb{E}|M - f(\boldsymbol{x})| \le \sigma_0/\sqrt{b}$.

Under Assumptions 3.1, 3.2, and 4.1, Lemma 3.3 gives, for $\eta_t \le \mu_{\mathcal{D}}/L_1$,

$$f(\boldsymbol{x}_{t+1}) \le f(\boldsymbol{x}_t) - \mu_{\mathcal{D}}\eta_t\|\nabla f(\boldsymbol{x}_t)\| + \frac{L_0}{2}\eta_t^2$$
$$+ 2\,\mathbb{E}\big[|M_t^+ - f(\boldsymbol{x}_t^+)| + |M_t^- - f(\boldsymbol{x}_t^-)|\big]. \quad (6)$$

By Assumption 4.1, $\mathbb{E}|M_t^{\pm} - f(\boldsymbol{x}_t^{\pm})| \le \sigma_0/\sqrt{b}$. Summing over $t$ and averaging gives:

**Theorem 4.2** (Average smoothness). *Let* $F_0 := f(\boldsymbol{x}_0) - f^\star < \infty$. *With* $\eta_t \equiv \eta \le \mu_{\mathcal{D}}/L_1$,

$$\frac{1}{T}\sum_{t=0}^{T-1}\mathbb{E}\|\nabla f(\boldsymbol{x}_t)\| = \frac{1}{\mu_{\mathcal{D}}}\mathcal{O}\Big(\frac{F_0}{\eta T} + L_0\,\eta + \frac{\sigma_0}{\eta\sqrt{b}}\Big).$$

*Optimizing* $\eta$ *yields*

$$\frac{1}{T}\sum_{t=0}^{T-1}\mathbb{E}\|\nabla f(\boldsymbol{x}_t)\| = \frac{1}{\mu_{\mathcal{D}}}\mathcal{O}\Big(\frac{L_1 F_0}{\mu_{\mathcal{D}}T} + \sqrt{\frac{L_0 F_0}{T}} + \frac{\sqrt{L_0\sigma_0}}{b^{1/4}}\Big).$$

To ensure $\varepsilon$-accuracy, that is, $\frac{1}{T}\sum_{t=0}^{T-1}\mathbb{E}\|\nabla f(\boldsymbol{x}_t)\| \le \varepsilon$, it suffices to pick

$$T = \mathcal{O}\Big(\frac{dL_1}{\varepsilon} + \frac{dL_0 F_0}{\varepsilon^2}\Big), \qquad b = \mathcal{O}\Big(\frac{(dL_0\sigma_0)^2}{\varepsilon^4}\Big),$$

using $\mu_{\mathcal{D}} \asymp 1/\sqrt{d}$. The total number of calls is

$$\text{CALLS} = 2bT = \mathcal{O}\left(\Big(\frac{dL_1}{\varepsilon} + \frac{dL_0 F_0}{\varepsilon^2}\Big)\cdot\frac{(dL_0\sigma_0)^2}{\varepsilon^4}\right).$$

The dominant term is therefore $\tilde{\mathcal{O}}(d^3\sigma_0^2/\varepsilon^6)$.

### 4.2. Case II: Standard sample smoothness

We now assume that each component function $f_\xi$ is $(L_0, L_1)$-smooth with respect to $\nabla f$:

$$\|\nabla f_\xi(\boldsymbol{x}) - \nabla f_\xi(\boldsymbol{y})\| \le (L_0 + L_1\|\nabla f(\boldsymbol{x})\|)\,\|\boldsymbol{x} - \boldsymbol{y}\|. \quad (7)$$

This is the *standard assumption* in the analysis of stochastic zeroth-order methods (see, e.g., (Ghadimi & Lan, 2013), which corresponds to the special case $L_1 = 0$). Our formulation is slightly more general.

We also assume bounded gradient variance:

**Assumption 4.3** (Bounded gradient variance). There exists $\sigma_1^2 < \infty$ such that for all $\boldsymbol{x} \in \mathbb{R}^d$, $\mathbb{E}\nabla f_\xi(\boldsymbol{x}) = \nabla f(\boldsymbol{x})$, and $\mathbb{E}\|\nabla f_\xi(\boldsymbol{x}) - \nabla f(\boldsymbol{x})\|^2 \le \sigma_1^2$.

Consequently, we show that for $M = \frac{1}{b}\sum_{i=1}^{b}\nabla f_{\xi_i}(\boldsymbol{x})$ is the average of an i.i.d minibatch of size $b$ independent of $\boldsymbol{s}|\boldsymbol{x}$, and $\boldsymbol{s} \sim \mathcal{D}$ isotropic (like the uniform distribution over the unit sphere, or normal distribution) we have $\mathbb{E}\big[|\langle M - \nabla f(\boldsymbol{x}), \boldsymbol{s}\rangle|\big] = \mathcal{O}\left(\sigma_1/\sqrt{db}\right)$.

Combining equation 7 with Assumption 4.3 and translation symmetry gives:

**Lemma 4.4** (Difference approximation error). *With* $\boldsymbol{x}_t^{\pm} = \boldsymbol{x}_t \pm \eta\boldsymbol{s}_t$ *and* $\mathbb{E}\|\boldsymbol{s}_t\|^2 = 1$,

$$\mathbb{E}\big|(M_t^+ - M_t^-) - (f(\boldsymbol{x}_t^+) - f(\boldsymbol{x}_t^-))\big|$$
$$\le 2(L_0 + L_1\|\nabla f(\boldsymbol{x}_t)\|)\eta^2 + \mathcal{O}\Big(\frac{\eta\sigma_1}{\sqrt{db}}\Big).$$

This yields the following bound:

**Theorem 4.5** (Sample smoothness). *Let* $F_0 := f(\boldsymbol{x}_0) - f^\star < \infty$. *Under Assumptions 3.2, 4.3, and sample smoothness, for* $\eta \le \mu_{\mathcal{D}}/(5L_1)$ *we have*

$$\frac{1}{T}\sum_{t=0}^{T-1}\mathbb{E}\|\nabla f(\boldsymbol{x}_t)\| = \frac{1}{\mu_{\mathcal{D}}}\mathcal{O}\left(\frac{F_0}{\eta T} + L_0\eta + \frac{\sigma_1}{\sqrt{db}}\right).$$

*Optimizing* $\eta$ *yields*

$$\frac{1}{T}\sum_{t=0}^{T-1}\mathbb{E}\|\nabla f(\boldsymbol{x}_t)\| = \frac{1}{\mu_{\mathcal{D}}}\mathcal{O}\left(\frac{L_1 F_0}{\mu_{\mathcal{D}} T} + \sqrt{\frac{L_0 F_0}{T}} + \frac{\sigma_1}{\sqrt{db}}\right).$$

To guarantee $\varepsilon$-accuracy, again, this means $\frac{1}{T}\sum_{t=0}^{T-1}\mathbb{E}\|\nabla f(\boldsymbol{x}_t)\| \le \varepsilon$, it suffices to pick

$$T = \mathcal{O}\left(\frac{dL_1}{\varepsilon} + \frac{dL_0 F_0}{\varepsilon^2}\right), \qquad b = \mathcal{O}\left(\frac{\sigma_1^2}{\varepsilon^2}\right).$$

The total complexity is

$$\text{CALLS} = 2bT = \mathcal{O}\left(\left(\frac{dL_1}{\varepsilon} + \frac{dL_0 F_0}{\varepsilon^2}\right)\cdot\frac{\sigma_1^2}{\varepsilon^2}\right).$$

The dominant term is therefore $\tilde{\mathcal{O}}(d\sigma_1^2/\varepsilon^4)$.

### 4.3. Significance of the results

We now compare the two regimes and highlight the improvements our analysis provides.

**Weak average smoothness.** In this regime we match the complexity $\tilde{\mathcal{O}}(d^3\sigma_0^2/\varepsilon^6)$ of (Boucherouite et al., 2024), but under a much weaker assumption: only the *average* function $f$ is required to be $(L_0, L_1)$-smooth, whereas they assumed each $f_\xi$ is globally $L$-smooth. Moreover, our analysis shows that the dominant term depends only on $L_0$, which can be much smaller than $L = L_0 + GL_1$ where $G$ bounds $\|\nabla f\|$.

**Standard sample smoothness.** Under the standard assumption that each $f_\xi$ is $(L_0, L_1)$-smooth with respect to $\nabla f$ (as in (Ghadimi & Lan, 2013), $L_1 = 0$), we obtain the improved complexity $\tilde{\mathcal{O}}(d\sigma_1^2/\varepsilon^4)$. Thus compared to the weak case, both the dependence on dimension and on accuracy improve. This shows that random search achieves the same scaling as gradient estimation methods, while avoiding discretization bias and remaining conceptually simpler.

In the next section, we show how variance reduction in the finite-sum setting can further improve the complexity when the dataset size $n$ is large.

## 5. Finite-Sum Case and Variance Reduction

We now consider the finite-sum setting

$$f(\boldsymbol{x}) = \frac{1}{n}\sum_{i=1}^{n} f_i(\boldsymbol{x}),$$

We assume here that each $f_i$ is $G$-Lipschitz, i.e, satisfies $\|\nabla f(\cdot)\| \le G$. We apply variance reduction by periodically evaluating the full dataset.

**Variance-reduced scheme.** Let $m$ be a fixed epoch length. Every $m$ iterations we perform a full pass over the dataset to compute exact values of the objective at the perturbed points. In between, we use stochastic minibatches of size $b$. Formally, the estimators are defined as

$$M_t^\pm = \begin{cases} f(\boldsymbol{x}_t^\pm), & \text{if } t \equiv 0 \pmod{m}, \\ \frac{1}{b}\sum_{j=1}^{b} f_{\xi_j}(\boldsymbol{x}_t^\pm), & \text{otherwise}, \end{cases}$$

where $\{\xi_j\}_{j=1}^{b}$ are sampled i.i.d. from $\{1, \ldots, n\}$.

**Translation-symmetric version.** Ordinarily, variance reduction methods require storing a *snapshot* $\tilde{\boldsymbol{x}}$ and using it to form a control variate correction for subsequent iterations. In our case, translation symmetry makes this correction automatic, since only the difference $M_t^+ - M_t^-$ matters. Thus we do not need to store $\tilde{\boldsymbol{x}}$ explicitly.

**Two-snapshot version.** For clarity, one can equivalently consider maintaining two separate snapshots $\tilde{\boldsymbol{x}}^+$ and $\tilde{\boldsymbol{x}}^-$, corresponding to the last time $\boldsymbol{x}_t^\pm$ was evaluated on the full dataset:

$$M_t^\pm = \begin{cases} f(\boldsymbol{x}_t^\pm), & \text{if } t \equiv 0 \pmod{m}, \quad \text{else} \\ f(\tilde{\boldsymbol{x}}^\pm) + \frac{1}{b}\sum_{j=1}^{b}\left(f_{\xi_j}(\boldsymbol{x}_t^\pm) - f_{\xi_j}(\tilde{\boldsymbol{x}}^\pm)\right). \end{cases}$$

This is the classical variance-reduced form, where the stochastic correction uses the difference between current and snapshot minibatches. It yields the same theoretical guarantees and total complexity as the translation-symmetric formulation, but requires storing two snapshots in memory. In contrast, the translation-symmetric construction avoids this storage burden.

**Error bound.** As in the sample smoothness regime, we analyze the difference $M_t^+ - M_t^-$. Using translation symmetry and the variance-reduction scheme, one can show

$$\mathbb{E}\left[\left|(M_t^+ - M_t^-) - (f(\boldsymbol{x}_t^+) - f(\boldsymbol{x}_t^-))\right|\right]$$
$$\le \mathcal{O}\left(\frac{\eta m}{\sqrt{b}}G\right), \tag{8}$$

where $\tilde{\boldsymbol{x}}_t$ denotes the most recent snapshot at which the function was fully evaluated.

**Averaged gradient bound.** Plugging this refined error into Lemma 3.3 then for all $\eta \le \mu_{\mathcal{D}}/L_1$, we obtain

$$\frac{1}{T} \sum_{t=0}^{T-1} \mathbb{E}\|\nabla f(\boldsymbol{x}_t)\| = \frac{1}{\mu_{\mathcal{D}}} \mathcal{O}\Big(\frac{F_0}{\eta T} + L_0 \eta + \frac{Gm}{\sqrt{b}}\Big). \quad (9)$$

Optimizing over $\eta$ yields

$$\frac{1}{T} \sum_{t=0}^{T-1} \mathbb{E}\|\nabla f(\boldsymbol{x}_t)\| = \frac{1}{\mu_{\mathcal{D}}} \mathcal{O}\Big(\sqrt{\frac{L_0 F_0}{T}} + \frac{L_1 F_0}{T} + \frac{Gm}{\sqrt{b}}\Big). \quad (10)$$

As before, $\mu_{\mathcal{D}} \asymp 1/\sqrt{d}$ for isotropic distributions.

**Parameter choices.** To ensure accuracy $\varepsilon$, it suffices to pick

$$T = \mathcal{O}\Big(\frac{dL_0 F_0}{\varepsilon^2} + \frac{dL_1 F_0}{\varepsilon}\Big), \quad b(m) = \mathcal{O}\Big(\frac{dm^2 G^2}{\varepsilon^2}\Big). \quad (11)$$

**Total complexity.** Each block of $m$ iterations requires

one full pass: $n + (m-1) \cdot b(m)$ stochastic calls.

Thus, over $T$ iterations the total number of function evaluations is

$$\text{CALLS}(m) = \frac{T}{m}\Big(n + (m-1)b(m)\Big).$$

Optimizing over $m$ gives the following leading complexity term:

$$\text{CALLS} = \mathcal{O}\Big(\frac{L_0 G^{2/3} F_0 \, d^{4/3} \, n^{2/3}}{\varepsilon^{8/3}}\Big). \quad (12)$$

**Discussion.** This rate is strictly better than the deterministic complexity $\mathcal{O}(dnL_0 F_0/\varepsilon^2)$ whenever $n \ge dG^2/\varepsilon^2$. Both the single-snapshot (translation-symmetric) and the two-snapshot variants achieve the same total complexity, but the former avoids the need to store snapshots in memory, which can be critical in large-scale settings. Variance reduction, therefore, provides a practical and theoretical advantage when the dataset size $n$ is large relative to the noise level, bringing the complexity closer to the optimal regime of stochastic optimization.

## 6. Learning with Helper or Human Feedback

In many applications, direct function evaluations may not be available or desirable, but one can obtain indirect feedback in the form of preferences or comparisons. Examples include:

- **A/B testing:** where we cannot measure the exact loss function, but can observe user preferences between two versions of a product or service (Kohavi et al., 2009).

- **Reinforcement learning with human feedback (RLHF):** where the reward signal is provided via human preference comparisons rather than explicit numerical scores (Christiano et al., 2017; Ouyang et al., 2022).

In such scenarios, our framework extends naturally by incorporating a *helper* (e.g., a human labeler, a proxy model, or a heuristic) that produces feedback about the relative quality of two states.

$\delta$-**inexact feedback.** We formalize the quality of the helper by requiring that for all $\boldsymbol{x}, \boldsymbol{y}$,

$$\mathbb{E}\big[\,\big|h(\boldsymbol{x}) - h(\boldsymbol{y}) - (f(\boldsymbol{x}) - f(\boldsymbol{y}))\big|\,\big] \le \delta,$$

that is, the helper provides a $\delta$-accurate approximation of the true difference in function values. We call this property $\delta$-*inexact feedback*. This is similar in spirit to the helper framework introduced in (Chayti & Karimireddy, 2024) for first-order methods and (Chayti et al., 2024) for second-order methods.

**Theoretical guarantee.** Using $\delta$-inexact feedback in place of exact evaluations in the stochastic three-point method, we can establish the following guarantee.

**Theorem 6.1** (Convergence with $\delta$-inexact feedback)**.** *Let $F_0 = f(\boldsymbol{x}_0) - f^\star < \infty$ and assume $(L_0, L_1)$-smoothness (Assumption 3.1) and directional distributional exploration (Assumption 3.2). With constant step size $\eta_t \equiv \eta \le \mu_{\mathcal{D}}/L_1$, we have*

$$\frac{1}{T} \sum_{t=0}^{T-1} \mathbb{E}\|\nabla f(\boldsymbol{x}_t)\| = \frac{1}{\mu_{\mathcal{D}}} \mathcal{O}\Big(\frac{F_0}{\eta T} + L_0 \, \eta + \frac{\delta}{\eta}\Big).$$

*Optimizing over $\eta$ yields*

$$\frac{1}{T} \sum_{t=0}^{T-1} \mathbb{E}\|\nabla f(\boldsymbol{x}_t)\| = \frac{1}{\mu_{\mathcal{D}}} \mathcal{O}\Big(\frac{L_1 F_0}{\mu_{\mathcal{D}} T} + \sqrt{\frac{L_0 F_0}{T}} + \sqrt{L_0 \delta}\Big).$$

**Accuracy guarantee.** By choosing

$$T = \mathcal{O}\Big(\frac{dL_1}{\varepsilon} + \frac{dL_0 F_0}{\varepsilon^2}\Big),$$

we ensure that

$$\frac{1}{T} \sum_{t=0}^{T-1} \mathbb{E}\|\nabla f(\boldsymbol{x}_t)\| \le \varepsilon + \mathcal{O}(\sqrt{d\delta}).$$

In other words, convergence can only be guaranteed up to a neighborhood of size $\mathcal{O}(\sqrt{d\delta})$, which reflects the intrinsic inexactness of the helper's feedback.

**Discussion.** This result shows that inexact comparison feedback degrades convergence only through the additive term $\sqrt{d}\delta$. When $\delta$ is small, the algorithm remains effective and achieves rates comparable to those with exact function evaluations. In practice, this provides theoretical support for random-search-based optimization in settings such as RLHF and A/B testing, where exact evaluations are unavailable but pairwise feedback is abundant.

## 7. On the Limitations of Momentum for Random Search

A natural question is whether the variance-reduction ideas developed for first-order stochastic optimization can be transferred to Random Search. In the first-order literature, several approaches exploit momentum to mitigate variance, including Heavy Ball (Polyak, 1964), momentum-based variance reduction such as STORM (Cutkosky & Orabona, 2019), and implicit gradient transport (Arnold et al., 2019). These methods help precisely when gradient noise is high and batches are small.

**Naive adaptation.** In Random Search we never access (nor approximate) gradients; the algorithm operates purely via function evaluations and their comparisons. It is tempting to adapt first-order momentum by reusing the same update forms but replacing stochastic gradients with function *differences*. For example, the Heavy Ball moving average $M_t = (1 - \beta)M_{t-1} + \beta \nabla f_\xi(\boldsymbol{x}_t)$ translates to

$$M_t = (1 - \beta)M_{t-1} + \beta\big(f_\xi(\boldsymbol{x}_t^+) - f_\xi(\boldsymbol{x}_t^-)\big),$$

and more elaborate momentum-based variance-reduction or implicit-transport schemes can be adapted analogously by plugging function differences in place of stochastic gradients.

**Negative result (why the classical analysis fails).** In first-order methods, momentum introduces a bias term, *but* the resulting variance reduction typically dominates, yielding net gains. In our comparison-based setting, that trade-off collapses: the stochastic term in the difference estimator scales as $\mathrm{Std}\big[f_\xi(\boldsymbol{x}_t^+) - f_\xi(\boldsymbol{x}_t^-)\big] \propto \eta$, because $\boldsymbol{x}_t^+$ and $\boldsymbol{x}_t^-$ are only $\Theta(\eta)$ apart. At the same time, the *signal* itself, $f(\boldsymbol{x}_t^+) - f(\boldsymbol{x}_t^-) \approx 2\eta\langle\nabla f(\boldsymbol{x}_t), \boldsymbol{s}_t\rangle$, also scales linearly with $\eta$. Consequently, the signal-to-noise ratio of the per-iteration difference is essentially *scale-invariant* in $\eta$, and exponential averaging (momentum) does not buy the same variance advantage as in the gradient case. Worse, the momentum recursion accumulates a *smoothing bias* that depends on how the differences $f(\boldsymbol{x}_t^+) - f(\boldsymbol{x}_t^-)$ drift across iterations; with random directions and non-stationary iterates, this bias does not shrink with $\eta$ at a rate that beats the $\Theta(\eta)$ stochastic term. The standard first-order proofs

that balance bias and variance, therefore break: reducing the variance via momentum no longer outpaces the induced bias, unless one resorts to *large minibatches*, which negates the intended benefit.

**Takeaway.** While momentum is a central tool for first-order methods, its direct transcription to Random Search—simply replacing gradients with a function differences—does not yield theoretical or practical improvements. The core obstruction is the step-size–dependent structure of both signal and noise in the difference oracle. Any meaningful momentum-like gain in this comparison-based regime will likely require *new* constructions (beyond literal translations of first-order schemata) that explicitly account for the $\Theta(\eta)$ scaling of both the signal and the noise.

## 8. Experiments

### 8.1. Logistic Regression Benchmark

**Setup.** We compare *Mi2P*, *RSGF*, and *ZO-CD* on a logistic-loss classification task using the Breast Cancer Wisconsin dataset. After a train/test split, the training set has $n = 455$ samples and $d = 30$ features. For $y_i \in \{\pm 1\}$, we optimize

$$f(\boldsymbol{x}) = \frac{1}{n}\sum_{i=1}^n \log\big(1 + \exp(-y_i\langle a_i, x\rangle)\big) + \frac{\lambda}{2n}\|x\|^2.$$

For batch sizes $b \in \{1, 5, 10, 25, 50, 100\}$ we run 20 trials each. Step sizes are tuned once per $b$ in a pilot phase, then fixed. All methods are compared under the same *function-query budget*, ensuring aligned $x$-axes in plots. We report $f(\boldsymbol{x})$ versus queries (mean $\pm$ one s.d. over runs).

**Methods.** Each iteration evaluates a minibatch objective $F_B$ with $|B| = b$:

- **RSGF:** sample $u$ on the unit sphere, set $g = \frac{F_B(x+\mu u) - F_B(x)}{\mu} u$, update $x \leftarrow x - \eta g$.
- **ZO-CD:** coordinate-wise two-point estimator $g = \sum_{i=1}^d \frac{F_B(x+\mu e_i) - F_B(x-\mu e_i)}{2\mu} e_i$, then $x \leftarrow x - \alpha g$.

Mi2P uses only the sign of $F_B(x+\alpha s) - F_B(x-\alpha s)$. Per iteration, Mi2P and RSGF cost $2b$ queries, ZO-CD costs $2bd$.

**Results.** Figure 1 shows $f(\boldsymbol{x})$ versus queries. For very small batches ($b = 1, 5$), Mi2P underperforms due to noisy comparisons that can flip the ordering of $x+\alpha s$ and $x-\alpha s$. As $b$ increases, comparisons become reliable: Mi2P matches or surpasses RSGF and consistently outperforms ZO-CD. This aligns with the theory that the misranking probability decreases with $b$, yielding more accurate updates.

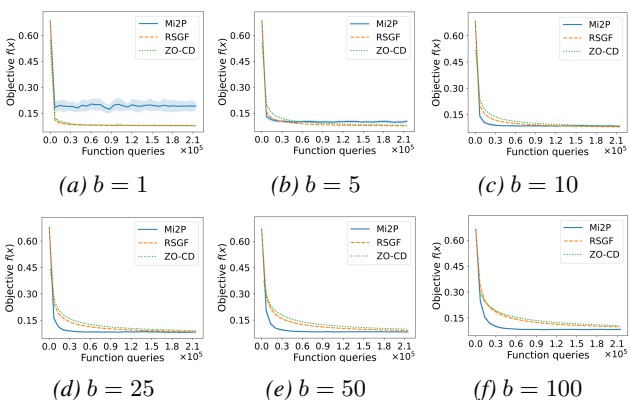

*(a) $b = 1$*     *(b) $b = 5$*     *(c) $b = 10$*

*(d) $b = 25$*     *(e) $b = 50$*     *(f) $b = 100$*

*Figure 1.* **Breast Cancer (logistic):** $f(\boldsymbol{x})$ vs. queries for different batch sizes. Mean $\pm$ one s.d. across 20 runs.

### 8.2. Policy-Search Experiments with Preference Feedback

We evaluate Mi2P, MiSTP, and Zeroth-Order Policy Gradient (ZPG) (Zhang & Ying, 2025) in a policy-search setting with pairwise preference feedback. ZPG applies the RSGF algorithm to a preference-based objective. For policies $\pi_\theta, \pi_{\theta'}$ with returns $R(\pi_\theta), R(\pi_{\theta'})$, we generate synthetic preferences via:

$$\pi_{\theta'} \succ \pi_\theta \iff \text{Bernoulli}\big(\sigma(R(\pi_{\theta'}) - R(\pi_\theta))\big) = 1,$$

where $\sigma(t) = 1/(1 + e^{-t})$ is the logistic link. Each comparison aggregates $N = 64$ trajectories.

We use three Gymnasium environments: `CartPole-v1`, `InvertedPendulum-v5`, and `Swimmer-v5`. All methods employ identical two-layer MLPs with 64 hidden units and tanh activations. Hyperparameters: ZPG uses smoothing $\mu = 10^{-1}$ and step size $\alpha = 10^{-3}$; Mi2P uses step size $10^{-1}$; MiSTP uses $2.5 \cdot 10^{-2}$. Step sizes were tuned on a small grid. We run each method for a fixed evaluation budget, reporting mean $\pm$ standard deviation of final returns over 10 seeds.

Figure 2 shows learning curves for all three environments. Mi2P and MiSTP consistently outperform ZPG, achieving $2\times$–$3\times$ higher mean returns on `CartPole-v1` and `Swimmer-v5` despite using identical policy architectures and trajectory budgets.

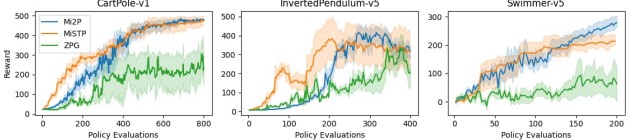

*Figure 2.* Learning curves for Mi2P, MiSTP, and ZPG on three benchmark environments. From left to right: `CartPole-v1`, `InvertedPendulum-v5`, and `Swimmer-v5`. Shaded regions show $\pm 1$ standard deviation over 10 random seeds.

*Table 1.* Final returns (mean $\pm$ std).

| Env | Mi2P | MiSTP | ZPG |
|---|---|---|---|
| CartPole | $\mathbf{473.9 \pm 19.7}$ | $464.1 \pm 63.4$ | $223.0 \pm 166.7$ |
| InvPend | $\mathbf{353.0 \pm 190.2}$ | $330.2 \pm 232.9$ | $257.9 \pm 214.6$ |
| Swimmer | $\mathbf{258.8 \pm 62.0}$ | $208.8 \pm 39.4$ | $71.4 \pm 78.2$ |

Table 1 summarizes final performance. Mi2P achieves the best performance across all environments, with MiSTP performing comparably on `CartPole-v1`. ZPG shows high variance and lower mean returns, particularly on `Swimmer-v5`.

## 9. Limitations, Future Work, and Conclusion

**Limitations.** Our analysis assumes $(L_0, L_1)$-smoothness and bounded variance. Extensions to weaker smoothness or heavy-tailed noise remain open. In addition, while momentum is highly effective in first-order methods, our negative result shows that its direct adaptation fails for Random Search, leaving variance reduction reliant on larger minibatches.

**Future work.** Promising directions include: (i) designing momentum-like schemes tailored to function differences, (ii) exploring more memory-efficient variance reduction methods, and (iii) evaluating Random Search in large-scale human-in-the-loop settings such as online A/B testing or RLHF.

**Conclusion.** We gave a unified analysis of Random Search under weak average smoothness, standard sample smoothness, and finite-sum objectives with variance reduction, and extended the framework to inexact human/helper feedback. Our results clarify the attainable rates of Random Search and highlight both its strengths and open challenges.

## Impact Statement

This paper presents work whose goal is to advance the field of Machine Learning. There are many potential societal consequences of our work, none of which we feel must be specifically highlighted here.

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

## A. Preliminaries

**Lemma A.1** (Variance of a minibatch average under bounded variance). *Let $(Z_i)_{i=1}^b$ be i.i.d. real-valued random variables with $\mathbb{E}[Z_1] = 0$ and $\mathrm{Var}(Z_1) \leq \sigma^2 < \infty$. Define the minibatch average $\overline{Z} := \frac{1}{b} \sum_{i=1}^b Z_i$. Then*

$$\mathbb{E}[\overline{Z}] = 0 \qquad and \qquad \mathrm{Var}(\overline{Z}) = \frac{\mathrm{Var}(Z_1)}{b} \leq \frac{\sigma^2}{b}.$$

*Proof.* Linearity of expectation gives $\mathbb{E}[\overline{Z}] = \frac{1}{b} \sum_{i=1}^b \mathbb{E}[Z_i] = 0$. For the variance,

$$\mathrm{Var}(\overline{Z}) = \mathrm{Var}\left(\frac{1}{b} \sum_{i=1}^b Z_i\right) = \frac{1}{b^2} \sum_{i=1}^b \mathrm{Var}(Z_i) + \frac{2}{b^2} \sum_{1 \leq i < j \leq b} \mathrm{Cov}(Z_i, Z_j).$$

Independence implies $\mathrm{Cov}(Z_i, Z_j) = 0$ for $i \neq j$, so $\mathrm{Var}(\overline{Z}) = \frac{1}{b^2} \cdot b\, \mathrm{Var}(Z_1) = \mathrm{Var}(Z_1)/b \leq \sigma^2/b$. $\qquad\square$

**Vector variant.** If $(\boldsymbol{z}_i)_{i=1}^b$ are i.i.d. $\mathbb{R}^d$-valued with $\mathbb{E}[\boldsymbol{z}_1] = \boldsymbol{0}$ and $\mathbb{E}\|\boldsymbol{z}_1\|^2 \leq \sigma^2$, then for $\overline{\boldsymbol{z}} := \frac{1}{b} \sum_{i=1}^b \boldsymbol{z}_i$,

$$\mathbb{E}[\overline{\boldsymbol{z}}] = \boldsymbol{0} \qquad and \qquad \mathbb{E}\|\overline{\boldsymbol{z}}\|^2 = \frac{1}{b} \mathbb{E}\|\boldsymbol{z}_1\|^2 \leq \frac{\sigma^2}{b}.$$

The proof is identical, using $\mathbb{E}\| \sum_i \boldsymbol{z}_i \|^2 = \sum_i \mathbb{E}\|\boldsymbol{z}_i\|^2$ by independence and zero mean.

**Lemma A.2** (Jensen's inequality). *Let $\phi : \mathbb{R} \to \mathbb{R}$ be convex and let $X$ be an integrable real-valued random variable. Then*

$$\phi(\mathbb{E}[X]) \leq \mathbb{E}[\phi(X)].$$

*In particular, for $\phi(x) = |x|$ we have $|\mathbb{E}[X]| \leq \mathbb{E}[|X|]$.*

*Proof.* By convexity, for any $x_0 \in \mathbb{R}$ there exists a subgradient $g \in \partial\phi(x_0)$ such that $\phi(x) \geq \phi(x_0) + g\,(x - x_0)$ for all $x \in \mathbb{R}$. Taking $x_0 = \mathbb{E}[X]$ and then expectations,

$$\mathbb{E}[\phi(X)] \geq \phi(\mathbb{E}[X]) + g\,\mathbb{E}[X - \mathbb{E}[X]] = \phi(\mathbb{E}[X]).$$

For $\phi(x) = x^2$, convexity gives $\mathbb{E}[|X|] \leq \sqrt{\mathbb{E}[X^2]}$ immediately. $\qquad\square$

**Lemma A.3** (Adaptive Smoothness Inequality). *Consider a differentiable function $f : \mathbb{R}^d \to \mathbb{R}$ satisfying the following property for all $\boldsymbol{x}, \boldsymbol{x}' \in \mathbb{R}^d$:*

$$\|\nabla f(\boldsymbol{x}) - \nabla f(\boldsymbol{x}')\| \leq \min\left(\mathcal{L}(\boldsymbol{x}), \mathcal{L}(\boldsymbol{x}')\right) \|\boldsymbol{x} - \boldsymbol{x}'\|,$$

*for some function $\mathcal{L} : \mathbb{R}^d \to \mathbb{R}^+$. Then, the function value is bounded by the quadratic approximation at $\boldsymbol{x}$ as follows:*

$$\left| f(\boldsymbol{x}') - f(\boldsymbol{x}) - \nabla f(\boldsymbol{x})^\top (\boldsymbol{x}' - \boldsymbol{x}) \right| \leq \frac{\mathcal{L}(\boldsymbol{x})}{2} \|\boldsymbol{x}' - \boldsymbol{x}\|^2.$$

*Proof.* We begin with the fundamental theorem of calculus for the function $f$:

$$f(\boldsymbol{x}') = f(\boldsymbol{x}) + \int_0^1 \nabla f(\boldsymbol{x} + t(\boldsymbol{x}' - \boldsymbol{x}))^\top (\boldsymbol{x}' - \boldsymbol{x})\,dt$$

$$= f(\boldsymbol{x}) + \nabla f(\boldsymbol{x})^\top (\boldsymbol{x}' - \boldsymbol{x}) + \int_0^1 \left[\nabla f(\boldsymbol{x} + t(\boldsymbol{x}' - \boldsymbol{x})) - \nabla f(\boldsymbol{x})\right]^\top (\boldsymbol{x}' - \boldsymbol{x})\,dt$$

Rearranging the terms, we isolate the error term:

$$f(\boldsymbol{x}') - f(\boldsymbol{x}) - \nabla f(\boldsymbol{x})^\top (\boldsymbol{x}' - \boldsymbol{x}) = \int_0^1 \left[\nabla f(\boldsymbol{x} + t(\boldsymbol{x}' - \boldsymbol{x})) - \nabla f(\boldsymbol{x})\right]^\top (\boldsymbol{x}' - \boldsymbol{x})\,dt$$

Taking the absolute value and applying the generalized Cauchy-Schwarz inequality, followed by the adaptive smoothness property:

$$\left|f(\boldsymbol{x}') - f(\boldsymbol{x}) - \nabla f(\boldsymbol{x})^\top (\boldsymbol{x}' - \boldsymbol{x})\right| \leq \left|\int_0^1 \left[\nabla f(\boldsymbol{x} + t(\boldsymbol{x}' - \boldsymbol{x})) - \nabla f(\boldsymbol{x})\right]^\top (\boldsymbol{x}' - \boldsymbol{x}) dt\right|$$

$$\leq \int_0^1 \left\|\nabla f(\boldsymbol{x} + t(\boldsymbol{x}' - \boldsymbol{x})) - \nabla f(\boldsymbol{x})\right\| \cdot \left\|\boldsymbol{x}' - \boldsymbol{x}\right\| dt$$

Using the adaptive smoothness property with $\boldsymbol{x}_t = \boldsymbol{x} + t(\boldsymbol{x}' - \boldsymbol{x})$ and $\boldsymbol{x}$:

$$\left\|\nabla f(\boldsymbol{x}_t) - \nabla f(\boldsymbol{x})\right\| \leq \min\left(\mathcal{L}(\boldsymbol{x}_t), \mathcal{L}(\boldsymbol{x})\right) \|\boldsymbol{x}_t - \boldsymbol{x}\|$$

Since $\min(\mathcal{L}(\boldsymbol{x}_t), \mathcal{L}(\boldsymbol{x})) \leq \mathcal{L}(\boldsymbol{x})$ and $\|\boldsymbol{x}_t - \boldsymbol{x}\| = \|t(\boldsymbol{x}' - \boldsymbol{x})\| = t\|\boldsymbol{x}' - \boldsymbol{x}\|$, we continue the inequality:

$$\left|f(\boldsymbol{x}') - f(\boldsymbol{x}) - \nabla f(\boldsymbol{x})^\top (\boldsymbol{x}' - \boldsymbol{x})\right| \leq \int_0^1 \left[\mathcal{L}(\boldsymbol{x}) \cdot t\|\boldsymbol{x}' - \boldsymbol{x}\|\right] \cdot \|\boldsymbol{x}' - \boldsymbol{x}\| dt$$

$$\leq \mathcal{L}(\boldsymbol{x})\|\boldsymbol{x}' - \boldsymbol{x}\|^2 \int_0^1 t\, dt$$

$$= \mathcal{L}(\boldsymbol{x})\|\boldsymbol{x}' - \boldsymbol{x}\|^2 \left[\frac{t^2}{2}\right]_0^1$$

$$= \frac{\mathcal{L}(\boldsymbol{x})}{2}\|\boldsymbol{x}' - \boldsymbol{x}\|^2.$$

$\square$

# B. Analysis of the General Algorithm

We consider the optimization problem

$$\min_{\boldsymbol{x} \in \mathbb{R}^d} f(\boldsymbol{x}).$$

At iteration $t$, Algorithm 1 samples a random direction $\boldsymbol{s}_t \sim \mathcal{D}$, defines two scalar quantities $M_t^{\pm}$, and updates

$$\boldsymbol{x}_{t+1} = \begin{cases} \boldsymbol{x}_t + \eta_t \boldsymbol{s}_t, & \text{if } M_t^+ \leq M_t^-, \\ \boldsymbol{x}_t - \eta_t \boldsymbol{s}_t, & \text{otherwise.} \end{cases}$$

Equivalently,

$$\boldsymbol{x}_{t+1} = \boldsymbol{x}_t - \eta_t \, \text{sign}(M_t^+ - M_t^-)\, \boldsymbol{s}_t.$$

We assume that $f$ is $(L_0, L_1)$-smooth, meaning that for all $\boldsymbol{x}, \boldsymbol{y} \in \mathbb{R}^d$,

$$\|\nabla f(\boldsymbol{x}) - \nabla f(\boldsymbol{y})\| \leq (L_0 + L_1\|\nabla f(\boldsymbol{x})\|)\|\boldsymbol{x} - \boldsymbol{y}\|. \tag{13}$$

By Lemma A.3, this implies that for all $\boldsymbol{x}, \boldsymbol{y} \in \mathbb{R}^d$,

$$f(\boldsymbol{y}) \leq f(\boldsymbol{x}) + \langle \nabla f(\boldsymbol{x}), \boldsymbol{y} - \boldsymbol{x}\rangle + \frac{L_0 + L_1\|\nabla f(\boldsymbol{x})\|}{2}\|\boldsymbol{y} - \boldsymbol{x}\|^2. \tag{14}$$

**Theorem B.1** (General Descent Inequality). *Let $f$ satisfy equation 13 and let the random directions $\boldsymbol{s}_t \sim \mathcal{D}$ satisfy Assumption 3.2, namely $\mathbb{E}\|\boldsymbol{s}_t\|^2 = 1$ and $\mathbb{E}[|\langle \nabla f(\boldsymbol{x}), \boldsymbol{s}_t\rangle|] \geq \mu_{\mathcal{D}}\|\nabla f(\boldsymbol{x})\|$. Then, for any step size $\eta_t \leq \mu_{\mathcal{D}}/L_1$, the update defined above satisfies*

$$\mathbb{E}[f(\boldsymbol{x}_{t+1})] \leq \mathbb{E}[f(\boldsymbol{x}_t)] - \tfrac{\mu_{\mathcal{D}}}{2}\eta_t \mathbb{E}\|\nabla f(\boldsymbol{x}_t)\| + \tfrac{L_0}{2}\eta_t^2 + 2\,\mathbb{E}\left[\left|M_t^+ - M_t^- - (f(\boldsymbol{x}_t^+) - f(\boldsymbol{x}_t^-))\right|\right], \tag{15}$$

*where $\boldsymbol{x}_t^{\pm} = \boldsymbol{x}_t \pm \eta_t \boldsymbol{s}_t$.*

*Proof.* Applying equation 14 with $\boldsymbol{y} = \boldsymbol{x}_t^{\pm} = \boldsymbol{x}_t \pm \eta_t \boldsymbol{s}_t$ and $\boldsymbol{x} = \boldsymbol{x}_t$ gives

$$f(\boldsymbol{x}_t^{\pm}) \leq f(\boldsymbol{x}_t) \pm \eta_t \langle \nabla f(\boldsymbol{x}_t), \boldsymbol{s}_t \rangle + \frac{L_0 + L_1 \|\nabla f(\boldsymbol{x}_t)\|}{2} \eta_t^2 \|\boldsymbol{s}_t\|^2.$$

Since $M_t^{\pm}$ are estimates of $f(\boldsymbol{x}_t^{\pm})$, we have

$$M_t^{\pm} \leq f(\boldsymbol{x}_t) \pm \eta_t \langle \nabla f(\boldsymbol{x}_t), \boldsymbol{s}_t \rangle + \frac{L_0 + L_1 \|\nabla f(\boldsymbol{x}_t)\|}{2} \eta_t^2 \|\boldsymbol{s}_t\|^2 + |M_t^{\pm} - f(\boldsymbol{x}_t^{\pm})|.$$

Hence,

$$M_t^{\pm} \leq f(\boldsymbol{x}_t) \pm \eta_t \langle \nabla f(\boldsymbol{x}_t), \boldsymbol{s}_t \rangle + \tfrac{L_0 + L_1 \|\nabla f(\boldsymbol{x}_t)\|}{2} \eta_t^2 \|\boldsymbol{s}_t\|^2 + \sum_{i \in \{\pm\}} |M_t^i - f(\boldsymbol{x}_t^i)|. \tag{16}$$

Suppose $M_t^+ \leq M_t^-$. Then $\boldsymbol{x}_{t+1} = \boldsymbol{x}_t^+$ and from equation 16,

$$M_t^+ \leq f(\boldsymbol{x}_t) - \eta_t |\langle \nabla f(\boldsymbol{x}_t), \boldsymbol{s}_t \rangle| + \tfrac{L_0 + L_1 \|\nabla f(\boldsymbol{x}_t)\|}{2} \eta_t^2 \|\boldsymbol{s}_t\|^2 + \sum_{i \in \{\pm\}} |M_t^i - f(\boldsymbol{x}_t^i)|.$$

Since $f(\boldsymbol{x}_{t+1}) = f(\boldsymbol{x}_t^+) \leq M_t^+ + |M_t^+ - f(\boldsymbol{x}_t^+)|$, we obtain

$$f(\boldsymbol{x}_{t+1}) \leq f(\boldsymbol{x}_t) - \eta_t |\langle \nabla f(\boldsymbol{x}_t), \boldsymbol{s}_t \rangle| + \tfrac{L_0 + L_1 \|\nabla f(\boldsymbol{x}_t)\|}{2} \eta_t^2 \|\boldsymbol{s}_t\|^2 + 2 \sum_{i \in \{\pm\}} |M_t^i - f(\boldsymbol{x}_t^i)|.$$

The same bound holds symmetrically when $M_t^- < M_t^+$.

Taking expectations and using $\mathbb{E}\|\boldsymbol{s}_t\|^2 = 1$ and $\mathbb{E}[|\langle \nabla f(\boldsymbol{x}_t), \boldsymbol{s}_t \rangle|] \geq \mu_{\mathcal{D}} \|\nabla f(\boldsymbol{x}_t)\|$, we get

$$\mathbb{E}[f(\boldsymbol{x}_{t+1})] \leq \mathbb{E}[f(\boldsymbol{x}_t)] - \eta_t \mu_{\mathcal{D}} \mathbb{E}\|\nabla f(\boldsymbol{x}_t)\| + \tfrac{L_0 + L_1 \mathbb{E}\|\nabla f(\boldsymbol{x}_t)\|}{2} \eta_t^2 + 2 \sum_{i \in \{\pm\}} \mathbb{E}|M_t^i - f(\boldsymbol{x}_t^i)|. \tag{17}$$

By the translation symmetry argument, shifting both $M_t^+$ and $M_t^-$ by any constant $C_t$ does not change the update rule. Thus, we can replace the last term by its optimal shift:

$$\min_{C_t} \sum_{i \in \{\pm\}} |M_t^i - C_t - f(\boldsymbol{x}_t^i)| = |M_t^+ - M_t^- - (f(\boldsymbol{x}_t^+) - f(\boldsymbol{x}_t^-))|.$$

Substituting into equation 17 and rearranging yields

$$\mathbb{E}[f(\boldsymbol{x}_{t+1})] \leq \mathbb{E}[f(\boldsymbol{x}_t)] - \eta_t \left( \mu_{\mathcal{D}} - \tfrac{L_1}{2} \eta_t \right) \mathbb{E}\|\nabla f(\boldsymbol{x}_t)\| + \tfrac{L_0}{2} \eta_t^2 + 2\mathbb{E}|M_t^+ - M_t^- - (f(\boldsymbol{x}_t^+) - f(\boldsymbol{x}_t^-))|.$$

Finally, since $\eta_t \leq \mu_{\mathcal{D}}/L_1$, we have $\mu_{\mathcal{D}} - \tfrac{L_1}{2} \eta_t \geq \mu_{\mathcal{D}}/2$, giving the claimed inequality equation 15. $\square$

## C. General Stochastic Functions with Subsampling

In this section, we consider the case where we define:

$$M_t^{\pm} := \frac{1}{b} \sum_{j=1}^{b} f_{\xi_j^t}(\boldsymbol{x}_t^{\pm}), \qquad \{\xi_j^t\}_{j=1}^{b} \text{ i.i.d.}$$

### C.1. Analysis under Average Smoothness

Suppose that $f$ satisfies the $(L_0, L_1)$-smoothness condition equation 13 and that the random directions $\boldsymbol{s}_t \sim \mathcal{D}$ satisfy Assumption 3.2, i.e. $\mathbb{E}[\|\boldsymbol{s}_t\|^2] = 1$ and $\mathbb{E}[|\langle \nabla f(\boldsymbol{x}_t), \boldsymbol{s}_t \rangle|] \geq \mu_{\mathcal{D}} \|\nabla f(\boldsymbol{x}_t)\|$.

Then, for any $\eta_t = \eta \leq \mu_{\mathcal{D}}/L_1$, we have

$$\mathbb{E}[f(\boldsymbol{x}_{t+1})] \leq \mathbb{E}[f(\boldsymbol{x}_t)] - \tfrac{\mu_{\mathcal{D}}}{2} \eta \, \mathbb{E}\|\nabla f(\boldsymbol{x}_t)\| + \tfrac{L_0}{2} \eta^2 + 2 \, \mathbb{E}\left[|M_t^+ - M_t^- - (f(\boldsymbol{x}_t^+) - f(\boldsymbol{x}_t^-))|\right]. \tag{18}$$

Let $X_j = f_{\xi_j^t}(\boldsymbol{x}_t^+) - f_{\xi_j^t}(\boldsymbol{x}_t^-) - (f(\boldsymbol{x}_t^+) - f(\boldsymbol{x}_t^-))$. Then $\mathbb{E}[X_j] = 0$ and suppose $\mathbb{E}[X_j^2] \leq \sigma_0^2 < \infty$.

Using the independence of $\xi_j^t$ and applying Jensen's inequality and Lemma A.1, it holds that

$$\mathbb{E}\big[\big|M_t^+ - M_t^- - (f(\boldsymbol{x}_t^+) - f(\boldsymbol{x}_t^-))\big|\big] = \mathbb{E}\left[\left|\frac{1}{b}\sum_{j=1}^b X_j\right|\right] \leq \sqrt{\mathbb{E}\left[\left(\frac{1}{b}\sum_{j=1}^b X_j\right)^2\right]} \leq \frac{\sigma_0}{\sqrt{b}}. \tag{19}$$

Substituting equation 19 into equation 18 yields

$$\mathbb{E}[f(\boldsymbol{x}_{t+1})] \leq \mathbb{E}[f(\boldsymbol{x}_t)] - \frac{\mu_{\mathcal{D}}}{2}\eta\,\mathbb{E}\|\nabla f(\boldsymbol{x}_t)\| + \frac{L_0}{2}\eta^2 + \frac{2\sigma_0}{\sqrt{b}}. \tag{20}$$

Averaging this inequality over $t = 0, \ldots, T-1$ and denoting $F_0 = f(\boldsymbol{x}_0) - f^\star$, with $f^\star = \inf_{\boldsymbol{x}} f(\boldsymbol{x}) > -\infty$, we obtain

$$\frac{1}{2T}\sum_{t=0}^{T-1}\mathbb{E}\|\nabla f(\boldsymbol{x}_t)\| \;\leq\; \frac{1}{\mu_{\mathcal{D}}}\left(\frac{F_0}{\eta T} + \frac{L_0}{2}\eta + \frac{2\sigma_0}{\eta\sqrt{b}}\right). \tag{21}$$

**Choice of stepsize.**   We choose the stepsize $\eta$ minimizing the right-hand side of equation 21 subject to $\eta \leq \frac{\mu_{\mathcal{D}}}{L_1}$, namely

$$\eta = \min\left(\frac{\mu_{\mathcal{D}}}{L_1},\; \sqrt{\frac{F_0}{L_0 T}},\; \sqrt{\frac{\sigma_0}{L_0\sqrt{b}}}\right). \tag{22}$$

Substituting this choice into equation 21 gives

$$\frac{1}{2T}\sum_{t=0}^{T-1}\mathbb{E}\|\nabla f(\boldsymbol{x}_t)\| \;=\; \frac{1}{\mu_{\mathcal{D}}}\,\mathcal{O}\left(\sqrt{\frac{L_0 F_0}{T}} + \frac{L_1 F_0}{\mu_{\mathcal{D}} T} + \frac{\sqrt{\sigma_0}}{b^{1/4}}\right). \tag{23}$$

**Complexity.**   To ensure $\frac{1}{2T}\sum_{t=0}^{T-1}\mathbb{E}\|\nabla f(\boldsymbol{x}_t)\| \leq \varepsilon$, we require

$$T = \mathcal{O}\left(\frac{L_0 F_0}{\mu_{\mathcal{D}}^2 \varepsilon^2} + \frac{L_1 F_0}{\mu_{\mathcal{D}}^2 \varepsilon}\right), \qquad b = \mathcal{O}\left(\frac{\sigma_0^2}{\mu_{\mathcal{D}}^4 \varepsilon^4}\right).$$

Since $\mu_{\mathcal{D}} = \mathcal{O}(d^{-1/2})$ for isotropic distributions, this simplifies to

$$T = \mathcal{O}\left(\frac{dL_0 F_0}{\varepsilon^2} + \frac{dL_1 F_0}{\varepsilon}\right), \qquad b = \mathcal{O}\left(\frac{d^2 \sigma_0^2}{\varepsilon^4}\right).$$

The total number of function evaluations (two per minibatch per iteration) is therefore of order

$$T \times b \;=\; \mathcal{O}\left(d^3 \sigma_0^2 F_0\left(\frac{L_0}{\varepsilon^6} + \frac{L_1}{\varepsilon^5}\right)\right). \tag{24}$$

## C.2. Analysis under individual Smoothness

For simplicity, in section 4.2, we assumed that each $f_\xi$ is $(L_0, L_1)$-smooth with respect to $\nabla f$. Here we show that the same improved rate still holds when each $f_\xi$ is instead $(L_0, L_1)$-smooth with respect to its own gradient $\nabla f_\xi$.

Assume that we work under the following assumptions:

**Assumption C.1.** For $\mathcal{P}$-a.e. $\xi$, the mapping $x \mapsto f_\xi(\boldsymbol{x})$ is $(L_0, L_1)$-smooth.

**Assumption C.2** (Bounded gradient variance)**.** There exists $\sigma \geq 0$ such that for all $\boldsymbol{x} \in \mathbb{R}^d$,

$$\mathbb{E}_\xi\big[\|\nabla f_\xi(\boldsymbol{x}) - \nabla f(\boldsymbol{x})\|^2\big] \;\leq\; \sigma^2.$$

**Lemma C.3** (Equivalent second-moment form)**.** *Assumption C.2 holds if and only if, for all $\boldsymbol{x} \in \mathbb{R}^d$,*

$$\mathbb{E}_\xi\big[\|\nabla f_\xi(\boldsymbol{x})\|^2\big] \;\leq\; \|\nabla f(\boldsymbol{x})\|^2 + \sigma^2.$$

*Proof.* Let $G_\xi(\boldsymbol{x}) := \nabla f_\xi(\boldsymbol{x})$ and $G(\boldsymbol{x}) := \nabla f(\boldsymbol{x}) = \mathbb{E}_\xi[G_\xi(\boldsymbol{x})]$. The variance decomposition identity gives

$$\mathbb{E}_\xi\big[\|G_\xi(\boldsymbol{x})\|^2\big] \;=\; \|G(\boldsymbol{x})\|^2 + \mathbb{E}_\xi\big[\|G_\xi(\boldsymbol{x}) - G(\boldsymbol{x})\|^2\big].$$

Therefore

$$\mathbb{E}_\xi\big[\|G_\xi(\boldsymbol{x}) - G(\boldsymbol{x})\|^2\big] \le \sigma^2 \quad\Longleftrightarrow\quad \mathbb{E}_\xi\big[\|G_\xi(\boldsymbol{x})\|^2\big] \le \|G(\boldsymbol{x})\|^2 + \sigma^2.$$

$\square$

**Lemma C.4.** *By sampling independent samples $\xi_1, \ldots, \xi_b$ from $\mathcal{P}$, we have for all $x, y \in \mathbb{R}^d$ :*

$$\mathbb{E}\|\frac{1}{b}\sum_{j=1}^{b}\nabla f_{\xi_j}(y) - \frac{1}{b}\sum_{j=1}^{b}\nabla f_{\xi_j}(x)\| \le \big(L_0 + L_1\sigma + L_1\|\nabla f(x)\|\big)\,\|x - y\|,$$

*and we have also:*

$$\|\nabla f(y) - \nabla f(x)\| \le \big(L_0 + L_1\sigma + L_1\|\nabla f(x)\|\big)\,\|x - y\|.$$

*Proof.* Let $x, y \in \mathbb{R}^d$. Using Assumption C.1, for $\mathbb{P}$-almost every $\xi$, we have:

$$\|\nabla f_\xi(y) - \nabla f_\xi(x)\| \;\le\; \big(L_0 + L_1\|\nabla f_\xi(x)\|\big)\,\|y - x\|.$$

By sampling independent samples $\xi_1, \ldots, \xi_b$ from $\mathcal{P}$, we have:

$$\|\frac{1}{b}\sum_{j=1}^{b}\nabla f_{\xi_j}(y) - \frac{1}{b}\sum_{j=1}^{b}\nabla f_{\xi_j}(x)\| \le \frac{\sum_{j=1}^{b}\|\nabla f_{\xi_j}(y) - \nabla f_{\xi_j}(x)\|}{b}$$

$$\le \left(L_0 + L_1\frac{\sum_{j=1}\|\nabla f_{\xi_j}(x)\|}{b}\right)\|x - y\|.$$

This implies that:

$$\mathbb{E}\|\frac{1}{b}\sum_{j=1}^{b}\nabla f_{\xi_j}(y) - \frac{1}{b}\sum_{j=1}^{b}\nabla f_{\xi_j}(x)\| \le \big(L_0 + L_1\mathbb{E}\|\nabla f_{\xi_1}(x)\|\big)\|x - y\|$$

$$\le L_0 + L_1\sqrt{\mathbb{E}\big(\|\nabla f_{\xi_1}(x)\|^2\big)}.\,\|x - y\|$$

$$\le \big(L_0 + L_1\sqrt{\sigma^2 + \|\nabla f(x)\|^2}\big)\|x - y\|$$

$$\le \big(L_0 + L_1\sigma + L_1\|\nabla f(x)\|\big)\|x - y\|.$$

For the second claim, using Jensen's inequality and assumption C.2, we obtain:

$$\|\nabla f(y) - \nabla f(x)\| = \big\|\mathbb{E}_\xi[\nabla f_\xi(y) - \nabla f_\xi(x)]\big\|$$

$$\le \mathbb{E}_\xi\big[\|\nabla f_\xi(y) - \nabla f_\xi(x)\|\big]$$

$$\le \mathbb{E}_\xi\big[L_0 + L_1\|\nabla f_\xi(x)\|\,\big]\|x - y\|$$

$$\le \big[L_0 + L_1\mathbb{E}\|\nabla f_\xi(x)\|\big]\|x - y\|$$

$$\le \big(L_0 + L_1\sqrt{\sigma^2 + \|\nabla f(x)\|^2}\big)\|x - y\|$$

$$\le (L_0 + L_1\sigma + L_1\|\nabla f(x)\|)\,\|x - y\|.$$

$\square$

For simplicity, we assume throughout the rest of this section that the search distribution $\mathcal{D}$ is uniform on the unit sphere.

**Lemma C.5.** *Let $u \in \mathbb{S}^{d-1}$ be uniformly distributed on the unit sphere and let $v \in \mathbb{R}^d$. Then*

$$\mathbb{E}\big[\langle v, u \rangle^2\big] = \frac{\|v\|^2}{d}.$$

*Proof.* Choose $Q$ an orthogonal matrix so that $Qv = \|v\| e_1$, where $e_1 = (1, 0, \ldots, 0)^\top$. Then

$$\mathbb{E}\big[\langle v, u \rangle^2\big] = \mathbb{E}\big[\langle Qv, Qu \rangle^2\big] = \|v\|^2 \, \mathbb{E}\big[\langle e_1, u \rangle^2\big] = \|v\|^2 \, \mathbb{E}[u_1^2] = \frac{\|v\|^2}{d}.$$

$\square$

By translation invariance (Section 3.5), we may shift $M_t^\pm$ by an optimal constant $c_t$ so that we are left with an estimation error given by:

$$\big|(M_t^+ - M_t^-) - (f(\boldsymbol{x}_t^+) - f(\boldsymbol{x}_t^-))\big|.$$

Under the assumptions stated above, the following lemma provides an explicit upper bound on the expected value of this error.

**Lemma C.6.** *Let $\boldsymbol{x}_t^\pm := \boldsymbol{x}_t \pm \eta s_t$, and assume Assumptions C.1 and C.2 hold. Suppose the search direction $s_t$ is drawn uniformly from the unit sphere $\mathbb{S}^{d-1}$. Let $\xi_{t,1}, \ldots, \xi_{t,b}$ be i.i.d. samples from $\mathcal{P}$, and define the mini-batch estimators*

$$M_t^\pm := \frac{1}{b} \sum_{j=1}^b f_{\xi_{t,j}}(\boldsymbol{x}_t^\pm).$$

*Then*

$$\mathbb{E}\big[\big|(M^+ - M^-) - (f(x_t^+) - f(x_t^-))\big| \,\big|\, x_t\big] \leq \frac{2\eta\sigma}{\sqrt{db}} + 2\eta^2\big(L_0 + L_1\sigma + L_1\|\nabla f(x_t)\|\big).$$

*Proof.* Denote: $x_t^\pm = x_t \pm \eta s_t$. By the fundamental theorem of calculus, we have:

$$\begin{cases} f_\xi(x_t^+) - f_\xi(x_t) = \int_0^{1} \langle \nabla f_\xi(x_t + u\eta s_t), \, \eta s_t \rangle \, du, \\ f_\xi(x_t^-) - f_\xi(x_t) = \int_0^{-1} \langle \nabla f_\xi(x_t + u\eta s_t), \, \eta s_t \rangle \, du. \end{cases} \implies f_\xi(x_t^+) - f_\xi(x_t^-) = \int_{-1}^{1} \langle \nabla f_\xi(x_t + u\eta s_t), \, \eta s_t \rangle \, du.$$

Averaging over the minibatch and subtracting the population identity gives:

$$(M^+ - M^-) - (f(x_t^+) - f(x_t^-)) = \eta \int_{-1}^{1} \Big\langle \underbrace{\frac{1}{b}\sum_{j=1}^b \nabla f_{\xi_j}(x_t + u\eta s_t) - \nabla f(x_t + u\eta s_t)}_{=: \, \Delta(x_t, s_t)}, \, s_t \Big\rangle \, du. \qquad (\star)$$

Denote: $\bar{g}(x) := \frac{1}{b}\sum_{j=1}^b \nabla f_{\xi_j}(x)$. We have:

$$\Delta(x_t, s_t) = \big(\bar{g}(x_t) - \nabla f(x_t)\big) + \big(\bar{g}(x_t + u\eta s_t) - \bar{g}(x_t)\big) - \big(\nabla f(x_t + u\eta s_t) - \nabla f(x_t)\big).$$

Plugging this into $(\star)$ and using the triangle inequality yields

$$\big|(M^+ - M^-) - (f(x_t^+) - f(x_t^-))\big| \leq 2\eta \big|\langle \bar{g}(x_t) - \nabla f(x_t), \, s_t \rangle\big|$$
$$+ \eta \int_{-1}^{1} \Big( \big|\langle \bar{g}(x_t + u\eta s_t) - \bar{g}(x_t), \, s_t \rangle\big|$$
$$+ \big|\langle \nabla f(x_t + u\eta s_t) - \nabla f(x_t), \, s_t \rangle\big| \Big) \, du. \qquad (\dagger)$$

By denoting $y_{u,t} = x_t + u\eta s_t$. By lemma C.4, we have:

$$\begin{cases} \big|\langle \nabla f(y_{u,t}) - \nabla f(x_t), \, s_t \rangle\big| \leq \|\nabla f(y_{u,t}) - \nabla f(x_t)\| \, \|s_t\| \leq \big(L_0 + L_1\sigma + L_1\|\nabla f(x_t)\|\big) |u|\eta \, \|s_t\|^2, \\ \\ \mathbb{E}\big[\big|\langle \bar{g}(y_{u,t}) - \bar{g}(x_t), \, s_t \rangle\big| \,\big|\, x_t, s_t\big] \leq \mathbb{E}[\|\bar{g}(y_{u,t}) - \bar{g}(x_t)\| \,|\, x_t, s_t]\|s_t\| \leq \big(L_0 + L_1\sigma + L_1\|\nabla f(x_t)\|\big) |u|\eta \, \|s_t\|^2, \end{cases}$$
$$(25)$$

Using Cauchy–Schwarz inequality and lemma C.5 we have:

$$\mathbb{E}\big[\big|\langle \overline{g}(x_t) - \nabla f(x_t),\, s_t\rangle\big|\, |x_t, \xi_1, \ldots, \xi_b\big] \leq \sqrt{\mathbb{E}[\langle \overline{g}(x_t) - \nabla f(x_t),\, s_t\rangle^2 \,|x_t, \xi_1, \ldots, \xi_b]}$$
$$= \frac{\|\overline{g}(x_t) - \nabla f(x_t)\|}{\sqrt{d}}$$

We have also: $\mathbb{E}\big[\|\overline{g}(x_t) - \nabla f(x_t)\| \,|x_t\big] \leq \sqrt{\mathbb{E}\big[\|\overline{g}(x_t) - \nabla f(x_t)\|^2 \,|x_t\big]} = \sqrt{\frac{1}{b}\mathbb{E}\big[\|\nabla f_{\xi_1}(x_t) - \nabla f(x_t)\|^2 \,|x_t\big]}$. This

implies that: $\mathbb{E}\big[\|\overline{g}(x_t) - \nabla f(x_t)\| \,|x_t\big] \leq \frac{\sigma}{\sqrt{b}}$. It follows that: $\mathbb{E}\big[\frac{\|\overline{g}(x_t) - \nabla f(x_t)\|}{\sqrt{d}} \,|x_t\big] \leq \frac{\sigma}{\sqrt{db}}$. Therefore:

$$\mathbb{E}\big[\big|\langle \overline{g}(x_t) - \nabla f(x_t),\, s_t\rangle\big|\, |x_t\big] \leq \frac{\sigma}{\sqrt{db}}.$$

Combining the bounds in 25 with (†) and the inequality above, we obtain:

$$\mathbb{E}\big[\big|(M^+ - M^-) - (f(x_t^+) - f(x_t^-))\big|\, |x_t\big] \leq \frac{2\eta\sigma}{\sqrt{db}} + 2\eta^2\big(L_0 + L_1\sigma + L_1\|\nabla f(x_t)\|\big).$$

$\square$

**Theorem C.7.** *Assume that assumptions C.1, and C.2 hold, the search directions sampled from the uniform distribution over the unit sphere, and the step size $\eta \leq \frac{\mu_{\mathcal{D}}}{16L_1}$. We have for all $t \geq 0$:*

$$\frac{1}{T}\sum_{t=0}^{T-1}\mathbb{E}\|\nabla f(\boldsymbol{x}_t)\| \;\leq\; \frac{4F_0}{\mu_{\mathcal{D}}\eta\,T} \;+\; \frac{16}{\mu_{\mathcal{D}}}\,\frac{\sigma}{\sqrt{db}} \;+\; \frac{18\eta}{\mu_{\mathcal{D}}}\,(L_0 + L_1\sigma).$$

*where $F_0 := f(\boldsymbol{x}_0) - f^\star$.*

*Proof of Theorem C.7.* Let $t \geq 0$. Using lemma C.4, $f$ is $(L_0 + L_1\sigma, L_1)$ smooth, and by applying Theorem B.1 , since $\eta \leq \mu_{\mathcal{D}}/L_1$, we have:

$$\mathbb{E}[f(\boldsymbol{x}_{t+1})] \leq \mathbb{E}[f(\boldsymbol{x}_t)] - \frac{\mu_{\mathcal{D}}}{2}\,\eta\,\mathbb{E}\|\nabla f(\boldsymbol{x}_t)\| + \frac{L_0 + L_1\sigma}{2}\,\eta^2 + 2\,\mathbb{E}\big[\big|M_t^+ - M_t^- - (f(\boldsymbol{x}_t^+) - f(\boldsymbol{x}_t^-))\big|\big].$$

Using Lemma C.6, we deduce that:

$$\mathbb{E}\big[f(\boldsymbol{x}_{t+1})\big] \leq \mathbb{E}\big[f(\boldsymbol{x}_t)\big] - \frac{\mu_{\mathcal{D}}}{2}\,\eta\,\mathbb{E}\big\|\nabla f(\boldsymbol{x}_t)\big\| + \frac{L_0 + L_1\sigma}{2}\,\eta^2 + \frac{4\eta}{\sqrt{db}}\,\sigma + 4\eta^2\Big(L_0 + L_1\sigma + L_1\,\mathbb{E}\big\|\nabla f(\boldsymbol{x}_t)\big\|\Big)$$

$$= \mathbb{E}\big[f(\boldsymbol{x}_t)\big] - \Big(\frac{\mu_{\mathcal{D}}}{2}\,\eta - 4\eta^2 L_1\Big)\mathbb{E}\big\|\nabla f(\boldsymbol{x}_t)\big\| + \underbrace{\Big(\frac{L_0 + L_1\sigma}{2}\,\eta^2 + \frac{4\eta}{\sqrt{db}}\,\sigma + 4\eta^2(L_0 + L_1\sigma)\Big)}_{=:\,\beta(\eta,b)}.$$

Let $\alpha(\eta, b) := \frac{\mu_{\mathcal{D}}}{2}\,\eta - 4\eta^2 L_1$. If we choose: $\eta \leq \frac{\mu_{\mathcal{D}}}{16L_1}$, then $\alpha(\eta, b) \geq \frac{\mu_{\mathcal{D}}}{4}\,\eta$. Summing the one–step bound over $t = 0, \ldots, T-1$ and telescoping gives:

$$\sum_{t=0}^{T-1}\alpha(\eta, b)\,\mathbb{E}\big\|\nabla f(\boldsymbol{x}_t)\big\| \;\leq\; \mathbb{E}\big[f(\boldsymbol{x}_0)\big] - \mathbb{E}\big[f(\boldsymbol{x}_T)\big] + T\,\beta(\eta, b) \;\leq\; F_0 + T\,\beta(\eta, b),$$

where $F_0 := f(\boldsymbol{x}_0) - f^\star$. Dividing by $T$ and by $\alpha(\eta, b) \geq \frac{\mu_{\mathcal{D}}}{4}\eta$ yields:

$$\frac{1}{T}\sum_{t=0}^{T-1}\mathbb{E}\big\|\nabla f(\boldsymbol{x}_t)\big\| \;\leq\; \frac{4F_0}{\mu_{\mathcal{D}}\eta\,T} \;+\; \frac{4}{\mu_{\mathcal{D}}\eta}\,\beta(\eta, b).$$

Expanding $\beta(\eta, b)$ and simplifying, we obtain:

$$\frac{1}{T}\sum_{t=0}^{T-1}\mathbb{E}\big\|\nabla f(\boldsymbol{x}_t)\big\| \;\leq\; \frac{4F_0}{\mu_{\mathcal{D}}\eta\,T} \;+\; \frac{2L_0 + 2L_1\sigma}{\mu_{\mathcal{D}}}\,\eta \;+\; \frac{16}{\mu_{\mathcal{D}}}\,\frac{\sigma}{\sqrt{db}} \;+\; \frac{16\eta}{\mu_{\mathcal{D}}}\,(L_0 + L_1\sigma).$$

$\square$

**Choice of stepsize.** We choose the stepsize $\eta$ minimizing the right-hand side of Theorem C.7 subject to $\eta \leq \frac{\mu_{\mathcal{D}}}{16L_1}$, namely

$$\eta \;=\; \min\!\left( \tfrac{\mu_{\mathcal{D}}}{16L_1}, \; \sqrt{\tfrac{F_0}{(L_0+L_1\sigma)\,T}} \right). \tag{26}$$

Substituting this choice into Theorem C.7 and simplifying yields

$$\frac{1}{T}\sum_{t=0}^{T-1} \mathbb{E}\|\nabla f(\boldsymbol{x}_t)\| \;=\; \frac{1}{\mu_{\mathcal{D}}}\,\mathcal{O}\!\left( \frac{L_1 F_0}{\mu_{\mathcal{D}} T} + \sqrt{\frac{(L_0+L_1\sigma)\,F_0}{T}} + \frac{\sigma}{\sqrt{d\,b}} \right). \tag{27}$$

**Complexity.** For the uniform distribution over the unit sphere, we have $\mu_{\mathcal{D}} \asymp d^{-1/2}$. With $T = \Theta\!\big(dL_1/\varepsilon + d(L_0 + L_1\sigma)F_0/\varepsilon^2\big)$ and $b = \Theta(\sigma^2/\varepsilon^2)$, we ensure $\frac{1}{T}\sum_{t=0}^{T-1} \mathbb{E}\|\nabla f(\boldsymbol{x}_t)\| \leq \varepsilon$. The total number of function evaluations is then

$$\text{CALLS} = 2bT \;=\; \mathcal{O}\!\left( \frac{d\,(L_0 + L_1\sigma)\,F_0\,\sigma^2}{\varepsilon^4} \right). $$

## D. Finite sum case and Variance reduction

We now consider the finite-sum setting under stronger regularity assumptions. Specifically, assume that $f = \frac{1}{n}\sum_{i=1}^n f_i$ is $(L_0, L_1)$-smooth and that each component function $f_i$ is $G$-Lipschitz, i.e.

$$|f_i(\boldsymbol{x}) - f_i(\boldsymbol{y})| \;\leq\; G\,\|\boldsymbol{x} - \boldsymbol{y}\|, \qquad \forall\,\boldsymbol{x}, \boldsymbol{y} \in \mathbb{R}^d. \tag{28}$$

We define

$$M_t^{\pm} = \begin{cases} f(\boldsymbol{x}_t^{\pm}), & \text{if } t \equiv 0 \pmod{m}, \\[2mm] \frac{1}{b}\sum_{j=1}^b f_{\xi_j}(\boldsymbol{x}_t^{\pm}), & \text{otherwise,} \end{cases}$$

**Theorem D.1** (Convergence under strong assumptions). *Under Assumption equation 28, suppose that $f$ is $(L_0, L_1)$-smooth, $\boldsymbol{s}_t \sim \mathcal{D}$ satisfies Assumption 3.2, and a variance-reduction scheme with snapshot period $m$ is used. Let $\eta_t \equiv \eta \leq \mu_{\mathcal{D}}/L_1$ and $F_0 := f(\boldsymbol{x}_0) - f^{\star} < \infty$. Then for each iteration,*

$$\mathbb{E}[f(\boldsymbol{x}_{t+1})] \;\leq\; \mathbb{E}[f(\boldsymbol{x}_t)] \;-\; \frac{\mu_{\mathcal{D}}}{2}\,\eta\,\mathbb{E}\|\nabla f(\boldsymbol{x}_t)\| \;+\; \frac{L_0}{2}\,\eta^2 \;+\; \frac{4\,G\,m}{\sqrt{b}}. \tag{29}$$

*Averaging over $t = 0, \ldots, T-1$, we obtain*

$$\frac{1}{T}\sum_{t=0}^{T-1} \mathbb{E}\|\nabla f(\boldsymbol{x}_t)\| \;\leq\; \frac{1}{\mu_{\mathcal{D}}} \left( \frac{F_0}{\eta T} + \frac{L_0}{2}\,\eta + \frac{4\,G\,m}{\sqrt{b}} \right). \tag{30}$$

Optimizing with respect to $\eta$ yields

$$\frac{1}{T}\sum_{t=0}^{T-1} \mathbb{E}\|\nabla f(\boldsymbol{x}_t)\| \;=\; \frac{1}{\mu_{\mathcal{D}}}\,\mathcal{O}\!\left( \frac{L_1 F_0}{\mu_{\mathcal{D}} T} + \sqrt{\frac{L_0 F_0}{T}} + \frac{G\,m}{\sqrt{b}} \right). \tag{31}$$

To guarantee $\frac{1}{T}\sum_{t=0}^{T-1} \mathbb{E}\|\nabla f(\boldsymbol{x}_t)\| \leq \varepsilon$, it suffices to choose

$$T = dF_0\,\mathcal{O}\!\left( \frac{L_1}{\varepsilon} + \frac{L_0}{\varepsilon^2} \right), \qquad b = b(\varepsilon, m) = \mathcal{O}\!\left( \frac{dG^2 m^2}{\varepsilon^2} \right). $$

The total number of function evaluations is then

$$\text{CALLS}(m) = \frac{T}{m}\big(n + (m-1)b(m)\big)$$

$$= \mathcal{O}\!\left( dF_0\Big(\frac{L_1}{\varepsilon} + \frac{L_0}{\varepsilon^2}\Big)\Big(\frac{n}{m} + \frac{dG^2 m^2}{\varepsilon^2}\Big) \right). \tag{32}$$

Minimizing over $m$ with $b(\varepsilon, m) \le n$ gives

$$m^\star = \min\left\{ \left(\frac{n\varepsilon^2}{dG^2}\right)^{1/3}, \left(\frac{n\varepsilon^2}{dG^2}\right)^{1/2} \right\},$$

and substituting $m^\star$ into equation 32 yields the optimal complexity

$$\text{CALLS}^\star = \mathcal{O}\left( \min\left\{ \frac{d^{4/3}n^{2/3}G^{2/3}}{\varepsilon^{2/3}}, dn \right\} F_0\left(\frac{L_1}{\varepsilon} + \frac{L_0}{\varepsilon^2}\right) \right). \tag{33}$$

equation 33 is better than the deterministic complexity whenever $n \ge \frac{dG^2}{\varepsilon^2}$. The quantity $dG^2$ can be interpreted as a noise level.

*Proof.* Define

$$X_j := f_{\xi_j^t}(\boldsymbol{x}_t^+) - f_{\xi_j^t}(\boldsymbol{x}_t^-) - (f(\boldsymbol{x}_t^+) - f(\boldsymbol{x}_t^-)).$$

By construction $\mathbb{E}[X_j] = 0$. Let $\tilde{\boldsymbol{x}}_t$ denote the snapshot point at the start of the current variance-reduction block of length $m$. Using the $G$-Lipschitz property of each $f_i$, we can write

$$\mathbb{E}[X_j^2] = \mathbb{E}\left[\left| f_{\xi_j^t}(\boldsymbol{x}_t^+) - f(\boldsymbol{x}_t^+) - \left(f_{\xi_j^t}(\boldsymbol{x}_t^-) - f(\boldsymbol{x}_t^-)\right) \right|^2\right]$$

$$= \mathbb{E}\left[\left| (f_{\xi_j^t}(\boldsymbol{x}_t^+) - f_{\xi_j^t}(\tilde{\boldsymbol{x}}_t)) - (f_{\xi_j^t}(\boldsymbol{x}_t^-) - f_{\xi_j^t}(\tilde{\boldsymbol{x}}_t)) - \left(f(\boldsymbol{x}_t^+) - f(\tilde{\boldsymbol{x}}_t)\right) + \left(f(\boldsymbol{x}_t^-) - f(\tilde{\boldsymbol{x}}_t)\right) \right|^2\right]$$

$$\le 2(2G)^2\left(\mathbb{E}\|\boldsymbol{x}_t^+ - \tilde{\boldsymbol{x}}_t\|^2 + \mathbb{E}\|\boldsymbol{x}_t^- - \tilde{\boldsymbol{x}}_t\|^2\right) \le (4Gm\eta)^2,$$

where the last inequality follows since the iterates remain within a distance $\mathcal{O}(m\eta)$ of the last snapshot over the $m$ local steps. Translation symmetry implies that this snapshot correction can be inserted without explicitly modifying the algorithm.

By independence of the samples $\xi_j^t$ and applying Lemma A.1 and Jensen's inequality, we have

$$\mathbb{E}\left[\left| M_t^+ - M_t^- - (f(\boldsymbol{x}_t^+) - f(\boldsymbol{x}_t^-)) \right|\right] = \mathbb{E}\left[\left| \frac{1}{b}\sum_{j=1}^b X_j \right|\right] \le \sqrt{\mathbb{E}\left[\left(\frac{1}{b}\sum_{j=1}^b X_j\right)^2\right]} \le \frac{4\eta Gm}{\sqrt{b}}. \tag{34}$$

Substituting equation 34 into Theorem B.1 directly yields the one-step bound equation 29. Averaging over $t$ and optimizing $\eta$ gives equation 30–equation 31. Finally, plugging these quantities into the total function-call count and minimizing over $m$ establishes the complexity bound equation 33. $\qquad\square$

# E. Learning with Helper or Human Feedback: Convergence Proof

We consider the setting where exact function evaluations are unavailable, and the algorithm receives *comparison feedback* from a helper (e.g., a human or a proxy). The helper returns a scalar signal $h(\boldsymbol{x})$ such that pairwise differences approximate true function differences.

**Assumption E.1** ($\delta$-inexact comparison feedback). There exists $\delta \ge 0$ such that for all $\boldsymbol{x}, \boldsymbol{y} \in \mathbb{R}^d$,

$$\mathbb{E}\left[ \left| h(\boldsymbol{x}) - h(\boldsymbol{y}) - (f(\boldsymbol{x}) - f(\boldsymbol{y})) \right| \right] \le \delta.$$

At iteration $t$, define the perturbed points $\boldsymbol{x}_t^\pm = \boldsymbol{x}_t \pm \eta_t \boldsymbol{s}_t$ and set the scalars

$$M_t^\pm := h(\boldsymbol{x}_t^\pm),$$

so that the update rule of Algorithm 1 becomes $\boldsymbol{x}_{t+1} = \boldsymbol{x}_t - \eta_t \,\text{sign}(M_t^+ - M_t^-)\,\boldsymbol{s}_t$.

**Theorem E.2** (Convergence with $\delta$-inexact feedback). *Suppose $f$ is $(L_0, L_1)$-smooth in the sense of equation 13, the random directions satisfy Assumption 3.2 ($\mathbb{E}\|\boldsymbol{s}_t\|^2 = 1$ and $\mathbb{E}[|\langle \nabla f(\boldsymbol{x}), \boldsymbol{s}_t \rangle|] \ge \mu_\mathcal{D}\|\nabla f(\boldsymbol{x})\|$), and the helper satisfies Assumption E.1. Let $F_0 := f(\boldsymbol{x}_0) - f^\star < \infty$ and choose a constant stepsize $\eta_t \equiv \eta \le \mu_\mathcal{D}/L_1$. Then*

$$\mathbb{E}[f(\boldsymbol{x}_{t+1})] \le \mathbb{E}[f(\boldsymbol{x}_t)] - \frac{\mu_\mathcal{D}}{2}\eta\,\mathbb{E}\|\nabla f(\boldsymbol{x}_t)\| + \frac{L_0}{2}\eta^2 + 2\delta. \tag{35}$$

*Averaging over $t = 0, \ldots, T - 1$ yields*

$$\frac{1}{T} \sum_{t=0}^{T-1} \mathbb{E}\|\nabla f(\boldsymbol{x}_t)\| \;\leq\; \frac{1}{\mu_{\mathcal{D}}} \left( \frac{F_0}{\eta T} + \frac{L_0}{2}\,\eta + \frac{2\delta}{\eta} \right). \tag{36}$$

Optimizing the right-hand side over $\eta$ gives

$$\frac{1}{T} \sum_{t=0}^{T-1} \mathbb{E}\|\nabla f(\boldsymbol{x}_t)\| \;=\; \frac{1}{\mu_{\mathcal{D}}}\,\mathcal{O}\!\left( \frac{L_1 F_0}{\mu_{\mathcal{D}} T} + \sqrt{\frac{L_0 F_0}{T}} + \sqrt{L_0\,\delta} \right). \tag{37}$$

In particular, choosing $T = \mathcal{O}\!\left( \frac{dL_1}{\varepsilon} + \frac{dL_0 F_0}{\varepsilon^2} \right)$ (with $\mu_{\mathcal{D}} \asymp d^{-1/2}$) ensures

$$\frac{1}{T} \sum_{t=0}^{T-1} \mathbb{E}\|\nabla f(\boldsymbol{x}_t)\| \;\leq\; \varepsilon \;+\; \mathcal{O}(\sqrt{d\delta}). \tag{38}$$

That is, convergence is guaranteed up to an $\mathcal{O}(\sqrt{d\delta})$ accuracy floor set by the inexactness of the helper.

*Proof.* Apply the general descent inequality (Theorem B.1) with $M_t^{\pm} = h(\boldsymbol{x}_t^{\pm})$:

$$\mathbb{E}[f(\boldsymbol{x}_{t+1})] \leq \mathbb{E}[f(\boldsymbol{x}_t)] - \tfrac{\mu_{\mathcal{D}}}{2}\,\eta\,\mathbb{E}\|\nabla f(\boldsymbol{x}_t)\| + \tfrac{L_0}{2}\,\eta^2 + 2\,\mathbb{E}\!\left[\left| \left(M_t^+ - M_t^-\right) - \left(f(\boldsymbol{x}_t^+) - f(\boldsymbol{x}_t^-)\right) \right|\right].$$

By Assumption E.1 with $(\boldsymbol{x}, \boldsymbol{y}) = (\boldsymbol{x}_t^+, \boldsymbol{x}_t^-)$, the last expectation is at most $2\delta$, giving equation 35. Summing from $t = 0$ to $T - 1$, taking total expectation, and telescoping yields

$$\mu_{\mathcal{D}}\,\eta \sum_{t=0}^{T-1} \mathbb{E}\|\nabla f(\boldsymbol{x}_t)\| \;\leq\; F_0 \;+\; \tfrac{L_0}{2}\,\eta^2 T \;+\; 2\delta\,T,$$

which rearranges to equation 36. Optimizing the right-hand side over $\eta$ under the constraint $\eta \leq \mu_{\mathcal{D}}/L_1$ (take $\eta = \min\{\mu_{\mathcal{D}}/L_1, \sqrt{F_0/(L_0 T)}, \sqrt{2\delta/L_0}\}$) gives equation 37. Finally, plugging $T = \mathcal{O}\!\left( \frac{dL_1}{\varepsilon} + \frac{dL_0 F_0}{\varepsilon^2} \right)$ (using $\mu_{\mathcal{D}} \asymp d^{-1/2}$ for isotropic $\mathcal{D}$) makes the first two terms $\leq \varepsilon$, and the remaining term is $\mathcal{O}(\sqrt{\delta})$, yielding equation 38. $\qquad\square$

# F. Failure of Classical Momenta

We now examine whether classical momentum techniques from first-order optimization can improve the performance of Random Search. We consider three natural extensions of popular momentum formulations: *Heavy-Ball* (Polyak, 1964), *momentum-based variance reduction (MVR)* (Cutkosky & Orabona, 2019), and *implicit gradient transport* (Arnold et al., 2019). Each method can be adapted to our comparison-based framework by replacing stochastic gradients with function differences.

**Momentum formulations.** Let $\boldsymbol{s}_t \sim \mathcal{D}$ and define $\boldsymbol{x}_t^{\pm} = \boldsymbol{x}_t \pm \eta \boldsymbol{s}_t$. We consider the following recursions:

$$\text{(Heavy-Ball)} \quad M_t = (1 - \beta)M_{t-1} + \beta\big(f_\xi(\boldsymbol{x}_t^+) - f_\xi(\boldsymbol{x}_t^-)\big),$$

$$\text{(MVR)} \quad M_t = (1 - \beta)\big(M_{t-1} + f_\xi(\boldsymbol{x}_t^+) - f_\xi(\boldsymbol{x}_t^-) + f_\xi(\boldsymbol{x}_{t-1}^+) - f_\xi(\boldsymbol{x}_{t-1}^-)\big) + \beta\big(f_\xi(\boldsymbol{x}_t^+) - f_\xi(\boldsymbol{x}_t^-)\big),$$

$$\text{(Transport)} \quad M_t = (1 - \beta)M_{t-1} + \beta\big(f_\xi(\tilde{\boldsymbol{x}}_t^+) - f_\xi(\tilde{\boldsymbol{x}}_t^-)\big), \quad \tilde{\boldsymbol{x}}_t^{\pm} = \boldsymbol{x}_t^{\pm} + \tfrac{1-\beta}{\beta}(\boldsymbol{x}_t^{\pm} - \boldsymbol{x}_{t-1}^{\pm}).$$

Each variant attempts to smooth the stochastic signal $f_\xi(\boldsymbol{x}_t^+) - f_\xi(\boldsymbol{x}_t^-)$ across iterations.

**Error decomposition.** For clarity, we focus on the Heavy-Ball recursion; the same reasoning applies to the others. Define the momentum error

$$e_t := M_t - \big(f(\boldsymbol{x}_t^+) - f(\boldsymbol{x}_t^-)\big),$$

and decompose the updates as

$$e_t = (1 - \beta)\big(e_{t-1} + b_t\big) + \beta v_t,$$

where

$$b_t := f(\boldsymbol{x}_t^+) - f(\boldsymbol{x}_t^-) - \big(f(\boldsymbol{x}_{t-1}^+) - f(\boldsymbol{x}_{t-1}^-)\big), \qquad v_t := \big(f_\xi(\boldsymbol{x}_t^+) - f_\xi(\boldsymbol{x}_t^-)\big) - \big(f(\boldsymbol{x}_t^+) - f(\boldsymbol{x}_t^-)\big).$$

**Bias–variance tradeoff.**  Assuming $f$ is $G$-Lipschitz, we have $|b_t| \leq 2G\eta$ and $\mathbb{E}[v_t^2] \leq (2G\eta)^2$. Taking expectations and unrolling the recursion gives

$$\mathbb{E}|e_t| \leq (1-\beta)^t \mathbb{E}|e_0| + (1-\beta)\sum_i (1-\beta)^{t-i}\mathbb{E}|b_i| + \beta \sqrt{\mathbb{E}\left[\left(\sum_i (1-\beta)^{t-i}v_i\right)^2\right]}$$

$$\leq (1-\beta)^t \mathbb{E}|e_0| + (1-\beta)\tfrac{2G\eta}{\beta} + 2G\eta\sqrt{\beta}.$$

Both the bias and variance terms scale linearly in $\eta$, and no choice of $\beta$ can improve this dependence. The optimal value is $\beta = 1$, which eliminates momentum altogether. The root cause is that the bias and variance of the difference estimator are of the same order in $\eta$, preventing the variance reduction mechanism from dominating as in first-order methods.

**Other variants.**  When each $f_\xi$ is $G$-Lipschitz, the same argument applies to the MVR recursion: both the correction and the variance term scale as $\mathcal{O}(G\eta)$, leaving no improvement over the non-momentum baseline.

The implicit transport momentum fails for a different reason. It requires extrapolated states $\tilde{\boldsymbol{x}}_t^\pm = \boldsymbol{x}_t^\pm + \tfrac{1-\beta}{\beta}(\boldsymbol{x}_t^\pm - \boldsymbol{x}_{t-1}^\pm)$, which introduces an additional displacement of order $\tfrac{1-\beta}{\beta}\eta$. Since the variance of the difference $f_\xi(\tilde{\boldsymbol{x}}_t^+) - f_\xi(\tilde{\boldsymbol{x}}_t^-)$ depends on the proximity of these states, this extrapolation *increases* variance by a factor of $\tfrac{1}{\beta}$. Consequently, the overall noise term scales as $\mathcal{O}(\tfrac{\sigma}{\sqrt{\beta}})$, offsetting the potential $\sqrt{\beta}$ gain from averaging. Once again, the best choice is $\beta = 1$, implying that not using momentum is preferable.

**Conclusion.**  In summary, all classical momenta—Heavy-Ball, momentum-based variance reduction, and implicit transport—fail to improve Random Search. The key difficulty lies in the structure of function-difference oracles: both the bias and the variance scale with the step size $\eta$, unlike in gradient-based methods where variance dominates. Consequently, naive momentum adaptation offers no benefit and can even increase noise.

