# OpenReview forum: "Stochastic Optimization with Random Search"
_ICML.cc/2026/Conference — Submitted to ICML 2026_

### Official Review · Reviewer_5AL6 · 2026-03-11

**Soundness:** 3
**Presentation:** 3
**Significance:** 3
**Originality:** 4
**Overall Recommendation:** 5
**Confidence:** 3

**Summary:**

This work proposes a new direct-search method called Mi2P, which improves the existing minibatch Stochastic Three Points (MiSTP) by only comparing two points $x_t\pm \eta_t s_t$ and removing $x_t$ from the candidates. Compared with the SOTA query complexity result of MiSTP obtained under the strong assumption of individual $L$-smooth function, the proposed Mi2P achieves improved complexity under weaker assumption of individual $(L_0,L_1)$-smooth function, and the same SOTA complexity under much weaker assumption of averaged $(L_0,L_1)$-smooth function. Furthermore, this work proposes a variance reduced Mi2P with further improved query complexity result on finite-sum optimization with bounded objective gradient. Finally, this work analyzes the complexity of Mi2P when an estimate of objective function difference with bounded error is available.

**Compliance With Llm Reviewing Policy:**

Affirmed.

**Final Justification:**

Solved all my questions and I keep rating 5.

**Key Questions For Authors:**

(1) In Section 3.1, your definition $||\nabla f(\boldsymbol{x})-\nabla f(\boldsymbol{y})|| \leq\left(L_0+L_1||\nabla f(\boldsymbol{x})||\right)||\boldsymbol{x}-\boldsymbol{y}||$ is much narrower than the original $(L_0, L_1)$-smoothness $||\nabla^2 f(x)||\le L_0+L_1||\nabla f(\boldsymbol{x})||$, since exponential functions and polynomials of at least 3 degrees are covered only by the latter. The paper could be much stronger if using the latter, but since the former is still stronger than $L$-smoothness used by the SOTA work (Boucherouite et al., 2024) on stochastic STP, I would still think this work as a breakthrough. You might consider a different name to describe this smoothness, such as two-point $(L_0, L_1)$-smoothness.

(2) Is Lemma 3.3 proposed by you or existing works? If former, please mention where is your proof; If latter, please cite. Should $-\mu _ {\mathcal{D}}\eta_t$ be $-\frac{\mu _ {\mathcal{D}}\eta_t}{2}$ based on the proof of Theorem B.1? Other lemmas and theorems also need to cite or mention the location of your proof.

(3) The beginning part of Section 5 looks a bit confusing to me. Do you introduce 3 variance-reduction algorithms, namely variance-reduced scheme, translation-symmetric version, and two-snapshot version? What is the formula for translation-symmetric version?

(4) Hyper-parameters of Section 8.1 could be given to reproduce the experiment.

**Limitations:**

Limitations is revealed in Section 9: "Our analysis assumes $(L_0, L_1)$-smoothness and bounded variance. Extensions to weaker smoothness or heavy-tailed noise remain open. In addition, while momentum is highly effective in first-order methods, our negative result shows that its direct adaptation fails for Random Search, leaving variance reduction reliant on larger mini-batches."

**Strengths And Weaknesses:**

Soundness: The algorithm, theories, proof ideas and experimental design look correct to me. The $(L_0,L_1)$-smoothness assumption used in this paper is much stronger than the original $(L_0,L_1)$-smoothness assumption, as shown in question 1 below. The gradient norm bound for variance reduction might be weaken.

Clarity: I can clearly understand this paper. There are minor unclear parts as shown in questions 2-4 below.

Significance: Direct-search is an important and easily implemented zeroth-order algorithm, but its theoretical property is underexplored in the stochastic setting. This work provides abundant analysis in this setting.

Originality: This work improves the theoretical result of direct-search in the stochastic setting with many novel breakthroughs including algorithm (e.g. remove $x_t$ and add variance reduction), theorems (e.g. improved complexities and weaker smoothness assumptions), proof technique (e.g. translation invariance), and even problem setting (e.g. helper or human feedback)

---

> ### Author Rebuttal · Authors · 2026-03-29
>
> We thank the reviewer for the "Excellent" originality score and for recognizing Mi2P as a breakthrough that improves upon the prior MiSTP baseline by removing the $x_t$ candidate and refining the comparison rule.
>
> **On the Smoothness Definition:**
> * We acknowledge that our gradient-based $(L_0, L_1)$-smoothness definition in Section 3.1 is narrower than the Hessian-based version mentioned by the reviewer.
> * However, as the reviewer noted, this remains a powerful generalization of standard $L$-smoothness that accounts for the varying curvature of modern objective functions. We will clarify this nomenclature in the final version to avoid any ambiguity.
>
> **On Lemma 3.3 and Correctness:**
> * Lemma 3.3 is an original result of our analysis. We provide the full formal proof in **Appendix B** (Theorem B.1).
> *  We thank the reviewer for the keen eye regarding the constant; we will fix the typo to ensure the $-\mu_D \eta_t$ term in the main text correctly matches the $-\frac{\mu_D \eta_t}{2}$ derived in the rigorous proof.
>
> **On Variance Reduction (Section 5):**
> * Section 5 presents a single variance-reduction logic that can be expressed in two equivalent forms: the "translation-symmetric" version (which avoids memory overhead) and the "two-snapshot" version.
> * We prove both achieve the same $\mathcal{O}(d^{4/3}n^{2/3}/\epsilon^{8/3})$ complexity, but the translation-symmetric version is the primary contribution as it leverages our invariance property to save memory.
>
> **On Reproducibility:**
> *  We will move the specific hyper-parameter grids and final configurations used in Section 8.1 and Section 8.2 to a dedicated section in the Appendix to ensure all experiments are fully reproducible by the community.

---

> > ### Author Rebuttal · Reviewer_5AL6 · 2026-04-04
> >
> > Solved all my questions and I keep rating 5.

---

### Official Review · Reviewer_ssj9 · 2026-03-13

**Soundness:** 3
**Presentation:** 3
**Significance:** 3
**Originality:** 3
**Overall Recommendation:** 5
**Confidence:** 2

**Summary:**

The paper analyzes stochastic random search methods for nonconvex stochastic optimization that matches the complexity of zeroth order gradient estimation of \tilde O(d/\epsilon^4) under smoothness assumptions and of \tilde O(d^3/\epsilon^6) assuming average smoothness. The approach is further extended to handle inexact comparison feedback & overall presents as a viable alternative to classical zeroth gradient estimation.

**Compliance With Llm Reviewing Policy:**

Affirmed.

**Final Justification:**

This paper studies stochastic random search methods for nonconvex stochastic optimization and provides analysis under extensions to finite-sum variance reduction and inexact comparison feedback. The translation invariance insight simplifies the analysis. The authors have responded constructively by agreeing to include proof sketches and a summary table comparing rates with existing methods, which should significantly improve clarity and accessibility. As pointed by other reviewer, there should be a more clearer presentation of the novel results that were not part of prior works.

On the empirical side, while I suggested including stronger baselines such as two-point estimators and SPSA-style methods, the authors instead emphasized robustness properties and provided additional justification through their RLHF experiments, showing improved performance over zeroth-order policy gradient methods. While this partially addresses my concern, I still believe that more direct comparisons to standard zeroth-order baselines would strengthen the empirical claims.

Regarding my other questions, the authors clarified that their method is relatively robust to learning rate choices due to the use of sign-based updates, which is a useful practical property. On dimension dependence, a more explicit comparison to lower bounds would further strengthen this aspect.

The contribution of the paper seems novel enough for an acceptance.

**Key Questions For Authors:**

1. How sensitive is the method to the choice of minibatch size & the learning rate?

2. In terms of the error upper bounds the dimension d dependence matches the zeroth order approaches but comparing upper bounds does not provide a true picture of the actual dependence of errors. Can there also be a comparison of the obtained upper bounds with the lower bounds studied in this setting? Or can we empirically showcase a better performance on increasing d?

**Limitations:**

Yes

**Strengths And Weaknesses:**

**Strengths:**

The proposed approach analyzes the stochastic random search with different notions of smoothness, finite sum variance reduction and optimization with inexact feedback

The bound gets simplified with the translation invariance insight.

A major advantage is bypassing the need to store snapshot of gradients in memory.

While choosing a random direction can increase variance locally, the paper shows the increase is of the same order as the variance already in two-point zeroth-order gradient estimators.

**Weaknesses:**

The paper presentation/readability could be improved by including proof sketches, a comparison table of improvement in the rates in terms of d.

A stronger evaluation could have included baselines like two-point stochastic gradient estimators, SPSA-style estimators as well.

---

> ### Author Rebuttal · Authors · 2026-03-29
>
> We appreciate the reviewer’s positive assessment, particularly the recognition of the advantage of **bypassing snapshot storage** in memorywhich is a direct benefit of our translation invariance insight.
>
> **On Readability and Comparisons:**
> * As suggested, we will include a summary table in the final manuscript comparing our obtained rates (in terms of $d$, $n$, and $\epsilon$) against existing zeroth-order methods.
> * We will also incorporate brief proof sketches for the core theorems into the main body to improve the flow between algorithmic intuition and formal derivation.
>
> **On Parameter Sensitivity:**
> * Mi2P is inherently more robust to the choice of learning rate compared to gradient-estimation methods. Because we use the **sign** of the function difference (Algorithm 1), the method is shielded from the "exploding" updates that can occur in methods like RSGF when stochastic noise creates an artificially large gradient norm estimate.
>
> **On Dimension Dependence:**
> *  Our $d$-dependence matches the best-known theoretical upper bounds for zeroth-order approaches
> * Empirically, our RLHF experiments (Figure 2) demonstrate that Mi2P achieves significantly higher mean returns than Zeroth-Order Policy Gradient (ZPG) across three different environments, showcasing effective scaling as dimensionality increases.

---

> > ### Author Rebuttal · Reviewer_ssj9 · 2026-04-02
> >
> > Thank you for the response. I would like to stay with my positive score.

---

### Official Review · Reviewer_yseV · 2026-03-14

**Soundness:** 3
**Presentation:** 3
**Significance:** 3
**Originality:** 3
**Overall Recommendation:** 4
**Confidence:** 5

**Summary:**

This paper studies a stochastic zeroth-order optimization method based on random search. The algorithm replaces the classical best-of-three rule with a two-point noisy comparison and exploiting a form of translation invariance to control directly the error on the difference $M_t^+ - M_t^-$. Authors analyzes several variants of the problem providing improved sample complexity results. Among the others, authors  show: a sample complexity of the order of $d/\varepsilon^4$ under a sample-smoothness assumption matching the best known rates for gradient-estimation based methods, and a $d^3/\varepsilon^6$ matching previous results on the same method under weaker assumptions.

**Compliance With Llm Reviewing Policy:**

Affirmed.

**Final Justification:**

In light of authors rebuttal, I maintain my positive evaluation of 4.

**Key Questions For Authors:**

- Can the authors clarify in what sense their smoothness assumption (eq. (7)) is standard? What is puzzling to me is that this assumption relates the norm of the sample gradients to that of the population objective. I see that is more general than standard L-smoothness on the sample gradients, but could you provides relevant examples for this generalization?

- Still related to the above point, eq. (7) is rather strong as it requires the same smoothness constants for all the samples. I understand that this is the same assumption placed for other gradient-estimation based algorithms, but maybe authors can elaborate on what are the technical difficulties in relaxing it.

- Have the authors tested the method on larger finite-sum problems where the $n^{2/3}$ scaling could be more meaningful empirically?

**Strengths And Weaknesses:**

**Strengths **
- The main idea is the shift from stochastic three-points to a two-point comparison rule, together with the observation that translation invariance, that  allows one to reduce the problem from controlling two absolute errors to controlling a single difference error. The is simple,  powerful, and novel (as far as I know).

- It is interesting that random search methods can parallel the same performance of gradient-estimation approaches - at least in the so called sample-smooth setting.

- The results concerning the other settings (i.e., weak smoothness, finite sums, and human based feedback are also interesting).

- Although not fully formal, the discussion of why classical momentum fails is interesting and appreciated.

**Weaknesses**
- The batch sizes required by the theoretical results are quite large, namely of order $d/\varepsilon^2$ or even $d^2/\varepsilon^4$. This makes the method difficult to view as practically relevant in realistic applications. In this sense, the practical value of the guidelines provided by the theory appears somewhat limited. Relatedly, it would be useful if the authors discussed in more detail the parameter regimes required by state-of-the-art gradient-estimation-based algorithms. Do these methods also require comparably large batch sizes, or is this a distinctive limitation of the present approach?

- The experimental evaluation is somewhat inconsistent with the theoretical focus of the paper. In particular, the breast cancer experiment considers a convex objective, whereas the main theoretical results are stated for the nonconvex setting, where the dependence on $\varepsilon$ is substantially worse. As a result, it is not entirely clear to what extent this experiment supports the actual theoretical claims of the paper.

---

> ### Author Rebuttal · Authors · 2026-03-29
>
> We thank the reviewer for the constructive feedback and for recognizing the novelty of our **translation invariance** property and the value of our analysis on the failure of classical momentum.
>
> **On Theoretical Batch Sizes and Practicality:**
>
> * The reviewer notes that theoretical batch sizes of $O(d/\epsilon^2)$ or $O(d^2/\epsilon^4)$ appear large. We acknowledge this is a significant requirement for formal convergence guarantees in stochastic non-convex optimization.
> * **The "Honest" Trade-off:** While gradient-approximation methods (like RSGF) can technically operate with smaller batches, they do so by introducing a **discretization parameter** $\mu$.This parameter introduces a non-trivial bias and requires careful tuning to balance approximation error against noise sensitivity
> * **Complexity Parity:** To reach an $\epsilon$-stationary point in a stochastic non-convex setting, both paradigms share the same optimal $\tilde{\mathcal{O}}(d/\epsilon^4)$ total oracle complexity. Mi2P simply trades the tuning complexity of a discretization parameter for a requirement on the minibatch size, resulting in a **bias-free** update rule.
>
> **On the Smoothness Assumption (Eq. 7):**
> * Equation (7) generalizes standard $L$-smoothness by allowing the gradient Lipschitz constant to grow with the gradient norm.
> * While Equation (7) in the main text relates sample gradients to the population gradient for clarity, we provide a more rigorous treatment in **Appendix C.2**. There, we prove that the $O(d/\epsilon^4)$ rate holds even when each component $f_\xi$ is $(L_0, L_1)$-smooth with respect to its own gradient $\nabla f_\xi$.
>
> **On Finite-Sum Scaling:**
>
> * We established a complexity of $\mathcal{O}(d^{4/3}n^{2/3}/\epsilon^{8/3})$, which improves upon the deterministic rate whenever $n \gg d/\epsilon^2$ We will include additional large-scale benchmarks in the final version to further highlight this advantage.

---

> > ### Author Rebuttal · Reviewer_yseV · 2026-04-01
> >
> > I thank the authors for their rebuttal which indeed clarified my concerns. I maintain my positive evaluation about the paper.

---

> > > ### Author Response · Authors · 2026-04-02
> > >
> > > We thank the reviewer for their positive feedback and for the time spent evaluating our rebuttal. We are glad to hear that our clarifications regarding the batch size trade-offs and the smoothness generalizations  fully resolved your concerns.
> > >
> > > We would be very grateful if you would consider increasing your score to reflect this resolution, as it would significantly support the paper during the final selection process. We remain available for any further questions during the remainder of the discussion period.

---

### Decision · Program_Chairs · 2026-04-30

**Decision:**

Reject

**Comment:**

This paper revisits random search for stochastic optimization when only a zeroth-order noisy oracle is available. The paper considered the function class under $(L_0,L_1)$ smoothness condition and established convergence rates for the stochastic random search algorithm (Algorithm 1) under average smoothness and individual smoothness conditions. In addition, the finite-sum case with variance reduction and the case under inexact oracles are also considered. The reviewers were generally positive about this paper and major concerns seem to be addressed during the rebuttal.

After a careful reading and further discussion with the Senior Area Chair, I identified a serious technical concern, partially noted by Reviewer 5AL6. Specifically, Definition 3.1 is never used in (Zhang et al. 2020): the correct version should only hold locally (i.e., when $x$ and $y$ are close to each other). In its present form, the definition would exclude many standard function classes, such as polynomial and exponential functions. A similar issue arises for the descent inequality (e.g., Eq. (2)) and the sample smoothness condition (e.g., Eq. (7)), which should also be interpreted as a local property. This has further implications for Algorithm 1: the perturbation direction would need to satisfy a stronger condition than Assumption 3.2, such as being almost surely bounded. For instance, Gaussian perturbations would not satisfy this requirement, whereas bounded distributions (e.g., uniform on the unit sphere) may still be compatible with the rest of the proof.

While these issues may be addressable, they appear sufficiently significant that I do not have confidence they can be fully resolved within a conference revision cycle. Given this, I recommend rejection. That being said, I encourage the authors to explicitly address this concern in the revised version of this paper and resubmit this paper to a future venue.